# Fast and versatile electrostatic disc microprinting for piezoelectric elements

Xuemu Li[1,2,3], Zhuomin Zhang [ID][1,2,3], Zehua Peng[1,2], Xiaodong Yan[1,2], Ying Hong [ID][1,2], Shiyuan Liu[1,2], Weikang Lin[1,2], Yao Shan[1,2], Yuanyi Wang[1] & Zhengbao Yang [ID][1,2] [✉]

Nanoparticles, films, and patterns are three critical piezoelectric elements with widespread applications in sensing, actuations, catalysis and energy harvesting. High productivity and large-area fabrication of these functional elements is still a significant challenge, let alone the control of their structures and feature sizes on various substrates. Here, we report a fast and versatile electrostatic disc microprinting, enabled by triggering the instability of liquid-air interface of inks. The printing process allows for fabricating lead zirconate titanate free-standing nanoparticles, films, and micro-patterns. The as-fabricated lead zirconate titanate films exhibit a high piezoelectric strain constant of 560 pm V$^{-1}$, one to two times higher than the state-of-the-art. The multiplexed tip jetting mode and the large layer-by-layer depositing area can translate into depositing speeds up to $10^9$ μm$^3$ s$^{-1}$, one order of magnitude faster than current techniques. Printing diversified functional materials, ranging from suspensions of dielectric ceramic and metal nanoparticles, to insulating polymers, to solutions of biological molecules, demonstrates the great potential of the electrostatic disc microprinting in electronics, biotechnology and beyond.

An ongoing surge in demand for microelectromechanical systems (MEMS), wearable/implantable electronics, miniaturized portable devices, and the Internet of Things has inspired the relentless pursuit of piezoelectric materials[1–3], thanks to their intrinsical property of coupling mechanical and electrical energy[4,5]. Films, micro-patterns, and nanoparticles are three significant piezoelectric elements for device-customized integration and efficient space utilization[6–9]. It has been a long sought-after goal to develop a high-productivity and large-area fabrication technique. Challenges of existing fabrication methods for piezoelectric films mainly include poor versatility, low volume production, high processing temperature, unsatisfying structural compactness, and prohibitively expensive apparatus[10–14]. Conventional micro-patterning techniques rely on screen printing[15] and photolithography/chemical etching[16,17]. However, these techniques often require high sintering temperatures (>1000 °C), complex and time-

consuming processing conditions, and hazardous materials. For example, patterning of poly(vinylidene fluoride) (PVDF) films needs the combination of spin coating and reactive ion etching[18,19] or micromold-assisted process[20] (soft lithography). For inorganic piezoelectric ceramics, the above patterning techniques lack compatibility with flexible and stretchable substrates. The foremost technical challenge for piezoelectric nanoparticle synthesis is particle agglomeration and customizable feature size[21]. Although some 3D printing techniques based on the ejection of ink from nozzles have been developed, most printing techniques still largely limited on the availability of ink, control over their microstructure and functionality, and printing speed[22–24].

In 1917, Zeleny[25] first observed that a strong electrostatic field could destabilize a microfluid interface separating drops from the surrounding air. The interface takes on a conical shape, now referred

[1]Department of Mechanical and Aerospace Engineering, Hong Kong University of Science and Technology, Clear Water Bay, Hong Kong, China. [2]Department of Mechanical Engineering, City University of Hong Kong, Hong Kong, China. [3]These authors contributed equally: Xuemu Li, Zhuomin Zhang. [✉]e-mail: zbyang@ust.hk

to as Taylor cone[26], and occurs when the microfluid is charged beyond the Rayleigh limit[27]. In a sufficiently intense electric field, either a string of droplets or a thin liquid jet that subsequently disintegrated into a large amount of droplets will be propelled from the cone's tip (Fig. 1a). Such electrostatically driven cone-jetting phenomena occur widely in nature and application, and two well-known examples are the ejection of streams of charged droplets from the tips of raindrops in a thunderstorm cloud and one immensely popular application for assaying large biomolecules: electrospray mass spectrometry. The Taylor cone can be formed from a microscopic pendent drop extruded from a round capillary, or macroscopic liquid films flowing to the tip[28] (Fig. 1b). Printing strategies inspired by such phenomena include electrospraying, electrospinning, and droplet focus printing[29–32], and they have good compatibility with complementary metal oxide semiconductor and MEMS fabrication techniques[33,34]. For the nozzle-based printing strategies, ink flow defined by the inner diameters of nozzles, as well as the capillary phenomena put an upper limit on the printing speed. Additionally, nozzles with smaller apertures suffer from

clogging and viscous losses, consequently, inks are limited to low-viscosity solutions free of large particles, which limits material versatility. To unleash the potential of high-speed and versatile printing allowed by electrostatic printing, some strategies based on multiple nozzle or nozzle-free have been proposed[35,36].

We develop an electrostatic disc microprinting (EDP) strategy to directly fabricate piezoelectric films, arbitrary micro-patterns and nanoparticles. The EDP process occurs via triggering the liquid-air interface instability of inks through an externally spiny thin disc. Benefiting from the large-volume printing process and in situ electrostatically induced modifications, the EDP process offers (i) on-demand printing strategies with a high depositing speed of $\sim10^9 \, \mu m^3 \, s^{-1}$, which is the fastest in the existing techniques for piezoelectric micrometer-thick films (<50 μm); (ii) lead zirconate titanate (PZT) films with excellent piezoelectric properties ($d_{33}$ of ~560 pm V$^{-1}$) outperforming the majority of previously reported PZT films; (iii) free-standing PZT nanoparticles in the size regime of 100 nm; (iv) PZT patterns with a fine feature size (minimum width is ~20 μm); and (v)

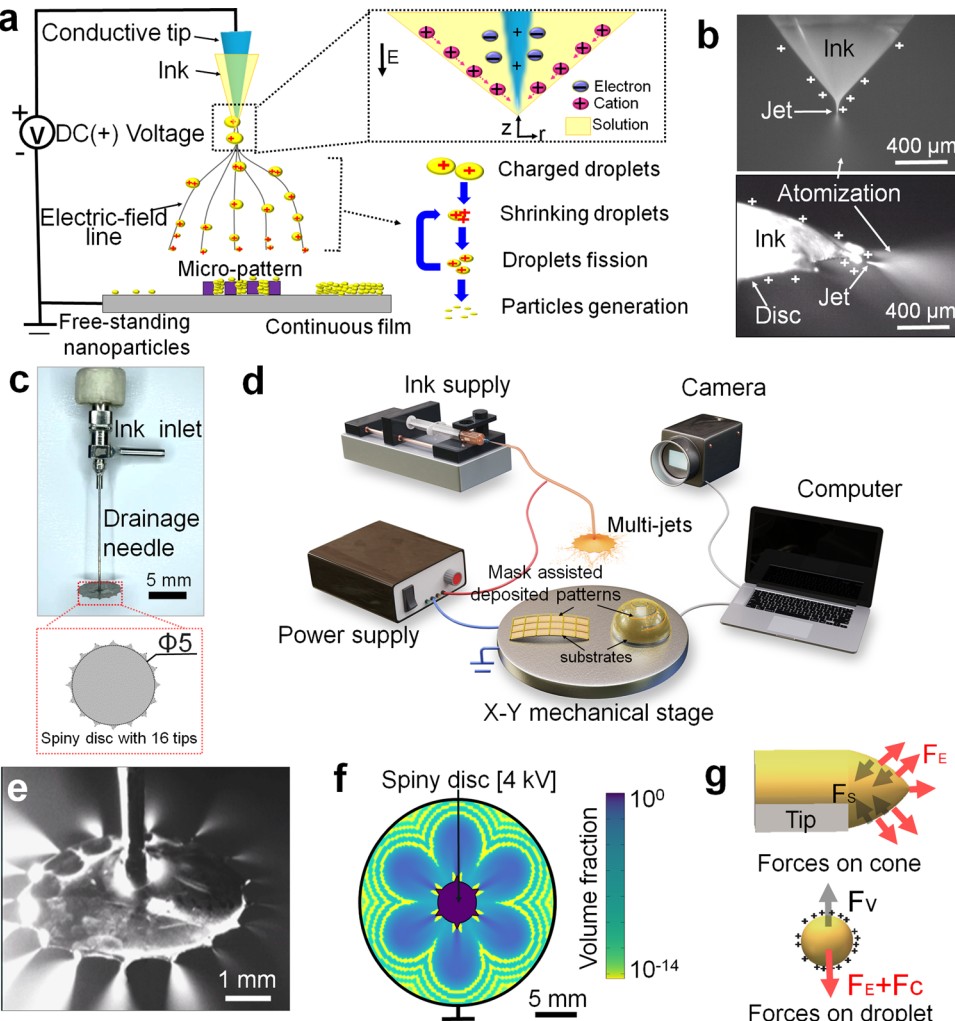

**Fig. 1 | Realization of the EDP. a** Schematic of cone-jetting formation from one tip. The electric field first distorts the initially spherical drop to a spheroidal shape and then leads to the formation of a cone at the ends. Later, tip streaming occurs, resulting in a small droplet disintegrating from the jet. The charged droplets will be guided along the electric-field line and broke into nanoparticles. **b** Optical photographs of the Taylor cone, jet and atomized droplets generated from a positively charged nozzle and conductive tips. **c** Optical photograph of inverted umbrella-shaped assembly of a drainage needle and a thin spiny disc with 16 tips. **d** Schematic of EDP printer. **e** High-speed movie capture of the EDP generated by applying ~5 kV

between the disc and the substrate. (Movie showing the EDP can be found in the supplementary material, Supplementary Movie 1). **f** Simulation of the fluid volume fraction around the thin spiny disc. The thin disc is equipped with six tips to simplify the simulation model. Additional simulation results can be found in the supplementary information, Fig. S3. **g** Competition between the two main forces (electrostatic force $F_E$ and surface tension $F_s$) on the Taylor cone apex. Forces acting on the atomized droplet. Electrostatic force $F_E$, Coulomb force $F_c$ and viscous force $F_v$.

printing capability for wide-ranging classes of materials such as dielectric ceramic, metal nanoparticles, insulating polymers, and biological molecules.

## Results

### Design of the EDP printer

The EDP printer basically comprise a home-made printing head (Fig. 1c), a syringe pump, an X-Y mechanical stage for mounting substrates, and a power supply connected between the spiny disc and the stage (Fig. 1d). The printing head is an inverted umbrella shaped assembly of a drainage needle and a thin spiny disc with 16 tips. The drainage needle with a diameter of 0.4 mm is connected to the syringe pump to supply inks. External drainage mode through this needle can effectively avoid clogging. The spiny disc has a thickness of 20 μm and a diameter of 5 mm (dedendum circle). The mechanical stage (with a diameter of 20 cm) is electrically-ground well in order to dissipate electrostatic charge of the substrates fast enough. Upon application of high voltage, the electric field is intensified effectively along the edge of the thin disc, especially at the tips due to the tip effect (Supplementary Note S1 and Supplementary Figs. S1, S2, and S3), which in turn induces the form of multiplexed jetting and tiny droplets (Fig. 1e, f). The moving trajectory of the atomized droplets is confined to a conical space. The underside of the conical space is the depositing area of the atomized droplets, and the deposited particles sizes depend on the depositing distance between the disc and the substrate, the ink properties, the applied voltage and the supply rate. The depositing distance is typically in the range of 1–50 mm. Shorter distances may not allow enough time for the droplets to dry up or cure sufficiently before reaching the substrate. In contrast, longer distances cause overly dispersed sediment and the corresponding porous films. As will be shown later, free-standing PZT nanoparticles can be obtained under long depositing distances. In this case, EDP can be used for nanoparticle fabrication. Films can be printed by continuous coalescence of the individual atomized droplets through a suitable combination of controlled moving speeds and the wetting of the substrate by inks. The thickness of the as-deposited films is typically related to the stage speeds and supply rate of inks. The printing process is fully controlled by a computer through the parameterization of the layer-by-layer deposition process to obtain the products with predefined microstructure, geometry and feature sizes.

Figure 1g shows the model for Taylor cone and the charged droplets. The Taylor cone is driven by the competition between electrostatic force $F_E$ and surface tension $F_s$ at the cone apex. The electrons of the fluid move to the wall of DC (+) voltage-attached tip, and causes the electric normal stress ($F_{E, n}$). The surface charges induced by charge conduction lead to electric tangential stress ($F_{E, t}$) along the surface meniscus with acceleration toward the cone apex (Fig. 1a). The electrostatic force and Coulomb force $F_c$ along the potential gradient combined with the synergistic effect from the viscous force $F_v$ allow the continued splitting of the charged droplets after solvent evaporation. In this process, the micro-droplets can gradually evolve into nano-spheres.

### Fabrication of piezoelectric films

Preparing inks with stable dispersion and reliable processability is one prerequisite to enable stable droplet atomization during EDP. Here, PZT films are fabricated using 20 h ball-milled PZT inks compositing commercial PZT nanoparticles (average diameter of 300 nm) with 0.6 M PZT sol. Their major physical properties are described in Table S1. In addition to the aforementioned ink properties, the process parameters are important for stable EDP process. The preferred voltage to form Taylor cone and thin jets during PZT ink depositing ranges from ~3.0 kV to ~10.0 kV. The average disc-substrate distance is ~5 mm, and the other optimized process parameters are listed in Table S2.

By exploiting the well-designed ink composition and processing optimization described above, multiplexed jets are formed (Supplementary Movie 1) and PZT films are fabricated. Morphological analysis of the as-deposited PZT films (5 depositing cycles) shows that the PZT particles are uniformly dispersed and interconnected well (Figs. S4 and S5a, b). PZT sols acted as structure-directing agent make individual particles well cross-linkage network. Infiltration using 0.4 M PZT sol through spin-coating is carried out to further improve the density of the as-deposited films (Fig. S5c). The surface and cross-sectional scanning electron microscopy (SEM) images show the well dense microstructures of the annealed PZT films with thickness ranging from ~6 to ~22 μm (Fig. 2a, b, c and Fig. S6). The demand for 3D conformal fabrication is increasingly growing, driven by the potential of integrating functional materials on a curvilinear surface for fundamentally new characteristic[37,38]. Considering that the height step of each deposited layer (<4 μm) is relatively small compared to the depositing distance, the effect of the reverse charging from as-deposited products to electric field is too small to disturb the precise layer-by-layer assembly. The deposited inks remain wet for a few seconds after printing, so the sediment will tend to spread to the recessed region of the substrates, which contribute to achieve 3D conformal printing. We use EDP to conformally and uniformly deposit PZT films on the 3D freeform stainless steel substrates, cloth substrates and wrinkled steel sheets (Fig. S7), demonstrating promising potential in 3D conformal electronics and smart textiles. The collision and dispersion of droplets on the substrate play an important role in the evolution of the as-deposited structures. For the high-concentration inks, atomized solvent-depleted particles tend to form a 'powdery' deposit and be exceedingly agglomerated owing to their poor mobility. The particle clusters will attract subsequent particles to agglomerate to them under the electrical field, referring to preferential landing. Extending the deposition time will favor the formation of larger and higher columns. Figure 2d, e shows the schematic diagram of this process and SEM micrograph of the columnar PZT structures, respectively. The dense/loose bilayer structure can be produced (Fig. 2f) by depositing of low-concentration inks and high-concentration inks in sequence.

Figure 2g shows the range of applied voltages for multiplexed jetting operation versus depositing distance for a flow rate equal to 10 μl min⁻¹. A large depositing distance results in an approximately linear increase in applied voltages. Optical microscopes (inset) demonstrate the tensile deformation and tip streaming of the liquid films in the electric field. Figure 2h shows a plot of the depositing speed as a function of the accessible piezoelectric films thickness for existing manufacturing techniques. The plot includes data from two kinds of piezoelectric films depositing techniques (Table S3): vapor phase depositions (including pulsed laser deposition and magnetron sputtering) and solution chemistry design-derived depositions (including sol-gel, composite sol-gel, aerosol deposition, electrophoretic deposition and EDP). As a general trend, the solution chemistry design-derived depositions can fabricate thicker films with a faster speed than vapor phase depositions. In this context, the EDP presented here is capable of depositing piezoelectric films with a thickness from ~1 μm to ~50 μm with the fabricating speed up to $10^9$ $μm^3$ $s^{-1}$. Please note that we include the time required for the intermittent steps, e.g., the pyrolyzation time of each precursor layer during spin-coating sol-gel process, into the fabricating speed calculations.

### Fabrication of micro-patterns and nanoparticles

Charged nanoparticles and droplets, generated by EDP, are simultaneously deposited onto the substrates, which are covered by the dielectric mask (Fig. S8). By keeping the substrate electrically-ground well, we electrostatically guide the nanoparticles towards the attractive area (holes of the dielectric mask or target region), whilst being away from the repulsive area (dielectric film or mask region) (Fig. 3a). Because the attractive area is recessed relative to the repulsive area,

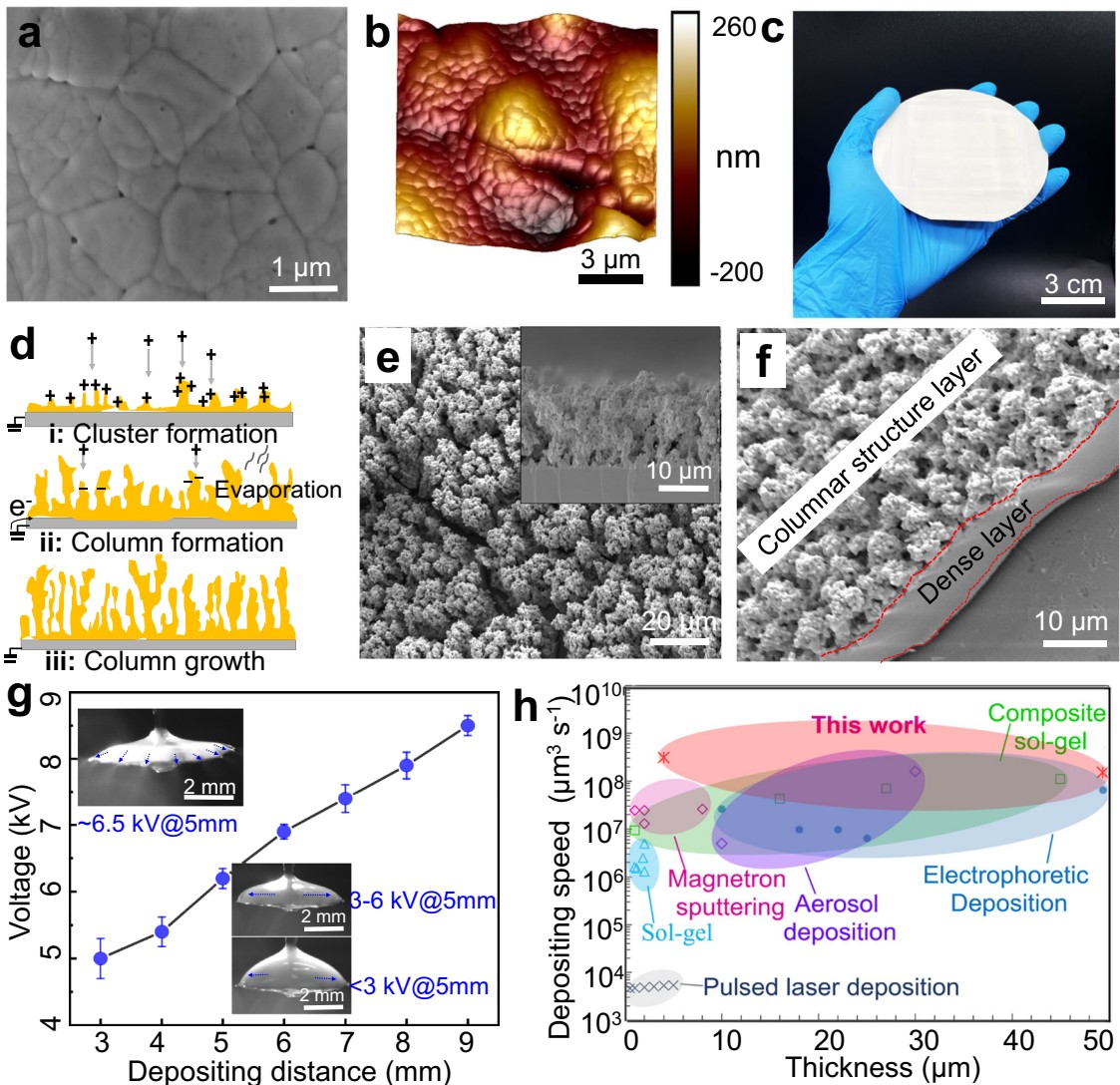

**Fig. 2 | Fabrication of piezoelectric films and the depositing speed. a** SEM image of surface topography of the annealed PZT film. **b** 3D AFM topography image of the annealed PZT film. **c** Optical photograph of PZT film deposited on a 4-inch Si wafer. **d** Schematic of the formation process for PZT columns. **e** SEM images showing the surface and cross-sectional topography of the PZT columnar structures. **f** SEM image showing the topography of the PZT dense/loose bilayer structure. **g** The range of applied voltages for multiplexed jetting operation versus depositing distance for the PZT ink. Optical microscopes (insets) demonstrate the tensile deformation and tip streaming of the liquid films under the action of an electric field. **h** Map of fabricating capabilities regarding manufacturing speed and the accessible film thickness (<50 μm).

charged particles accumulation around the edge of each hole (mask region) occurs and will prevent nanoparticles deposition near the edge of the attractive area. This funneling effect will reduce the width of the area coated with deposited nanoparticles, which can push the resolution of assembly beyond that of the defined dielectric mask. The mask can be easily removed with tweezers following deposition and heat treatment without affecting the functionality of the patterns. Examples of EDP patterns include arrays of linear structures, circular structures, squares and some complex patterns (Fig. 3b, c and Fig. S9). The feature size of the printed structures mainly depends on the dimensions of the dielectric mask, charged droplet diameter, and the wettability of the ink on the substrates. The increase of thickness is proportional to the number of layers. We miniaturize the feature size of the printed PZT line to ~20 μm (thickness to width ratio of 0.5) (Fig. 3d). For silver inks, the minimum width of ~10 μm (thickness to width ratio of ~1) is obtained. The sloping side walls for deposited patterns are formed due to the funneling effect and the diffusion of wet films (Fig. S10). Given that the mask will be detached after the deposition and heat treatment, it is believed that a significant amount

of unused deposited materials would be wasted. For example, using polyimide mask with ~200 μm-wide grooves, the coverage ratio of PZT on the target region and mask region is 92.5 ± 1.2% and 21.6 ± 6.6%, respectively. A sparse amount of PZT is deposited on the mask region due to the unstable jet and the electrostatic repulsion between the deposited and coming droplets. The particle source used provides the flexibility to use different materials, enabling the printing of multi-material patterns. Patterned structures can be deposited on flexible planar substrates and 3D free-form substrates (Fig. 3e), demonstrating the esthetic diversity of the EDP patterns. The printed letters 'Hello world' composed of different materials (silver and PZT) (Figs. 3e and S9a) demonstrate the material versatility of EDP. The patterns are directly deposited onto the polyethylene terephthalate (PET) substrates. The sediment thickness (within 100 μm) and area (within 5 × 5 cm) are far smaller than the depositing distance (>2 mm) and the area of electrically-ground stage (with a diameter of 20 cm), respectively. Therefore, the substrates and dielectric masks have very limited effect on the electric field between disc and ground electrode. The stable atomizing state is maintained throughout the entire deposition

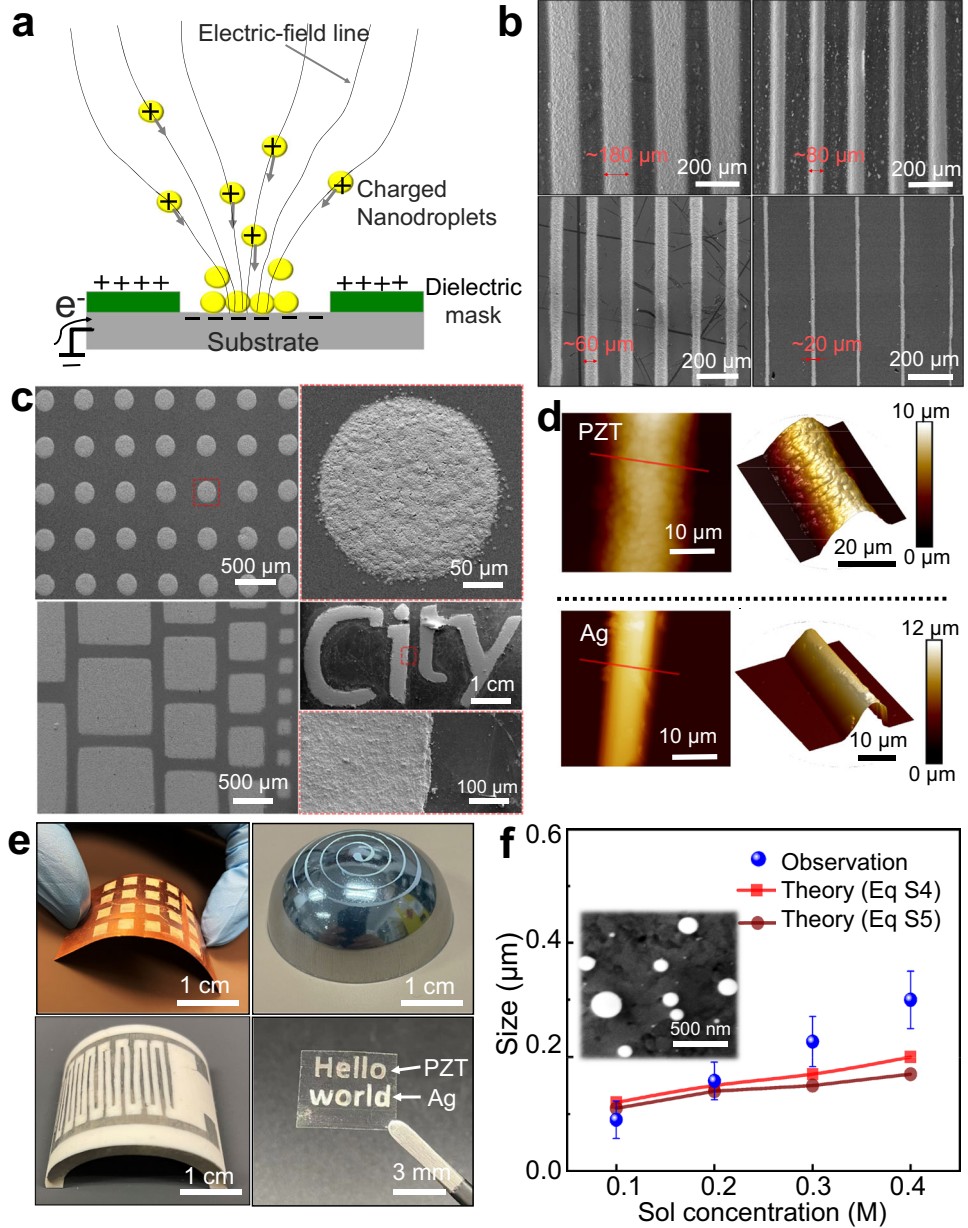

**Fig. 3 | Fabrication of micro-patterns and nanoparticles. a** Schematic of the process for printing micro-patterns through a hole-containing dielectric mask. **b** SEM images of PZT linear structures arrays with different feature sizes. The linear structures with a width of ~180 μm and ~20 μm are fabricated via polyimide mask with ~200 μm-wide and ~80 μm-wide grooves, respectively. **c** SEM images of the printed PZT arrays of circular structures, squares and printed micro-letters ("City") on Si substrates. **d** 3D AFM topography image of the printed PZT and silver lines. **e** Optical photographs of micro-patterns deposited on copper foil, steel hemisphere, alumina curve surface, and PET substrate. **f** Diameter variation of the PZT particles as a function of sol concentrations. The inset shows the SEM image of free-standing PZT particles with different diameters. Additional results can be found in the supplementary information, Fig. S11.

process. The PZT patterns deposited on the PET substrates are only dried at 200 °C for 2 min.

In the EDP process, the inevitable solvent evaporation phenomenon determines the geometrical dimensions of the atomized elements (Fig. 1a). Once atomized, the structural integrity of droplets is dependent upon the competition of surface tension with the electrostatic repulsion. Up to a point known as the Rayleigh limit, droplet fragmentation will be constrained where surface tension holds the repulsive forces in check. Due to the solvent evaporation, however, continuous shrinkage of droplets gradually bring the charges closer together, increasing repulsion proportionally. Eventually, the Rayleigh limit is overcome and the droplets undergo Coulombic explosion, splitting into progeny droplets in which the process is reset. The solvent evaporation rate depends on multiple

parameters, mainly including ink physical properties, travel distance of droplets, and the diameter of the pendant drop. The ink physical properties, which particularly depends on ink concentration, has a great impact on the atomization process and atomized particles sizes, as noted in Note S2. Figure 3f and Fig. S11 show the deposited PZT particles diameters as a function of inks concentration. As the ink concentration is increased, the particle diameter is increased. This result is expected, taking into account the self-dispersion of the electrified droplets and their volumetric shrinkage. The formation of steady atomization and control of temperature near atomizing area during EDP can one-step fabricate PZT particles demonstrating monodispersity and smooth morphologies (Inset of Fig. 3f). The optimized parameters for one-step depositing PZT nanoparticles are listed in Table S4.

## Piezoelectric performances of films

X-ray diffraction (XRD) spectrum show the crystallographic structures of EDP PZT films and dip-coated PZT films (Fig. 4a). Clearly both of the annealed films have crystallized in a pure perovskite phase with a (011)-preferred orientation and no evidence of secondary phase formation, such as the pyrochlore phase, is detected. The relative intensities of the XRD peaks from 42.5° to 46.5° fitted by Gauss function show their phase features of coexistence of crystal plane (002)R, (002)T and (020)T (Fig. 4b), which corresponds to the rhombohedral phase and the tetragonal phase, respectively. The crystallographic refinement results further confirm that MPB PZT films composed of R phase and T phase are synthesized (Fig. S12). Their lattice parameters are summarized in Table S5. Compared with those of the dip-coated PZT films, all lattice constants ($a_R$, $a_T$, and $c_T$) of EDP PZT films have increased, and the level of $c_T/a_T$ increase. The rhombohedral distortion, 90°-$\alpha_R$ also increases from 0.242° to 0.351°. These structural distortions (Fig. 4c) are caused by in-situ electric field[39,40], and will contribute the improvement of ferroelectric and piezoelectric properties[41].

Piezoelectric properties of the PZT films are studied via piezo-response force microscopy (PFM), which gives the voltage-induced deformation and discloses the domain structure. As shown in the amplitude and phase images (Fig. 4d, e), the color is virtually uniform inside the grains, suggesting that they exist in a single domain state. The mean effective piezoelectric coefficient $d_{33}$ of 560 pm V$^{-1}$ is observed for <10 μm thick PZT films (Fig. 4f). The variation of the relative dielectric constant $\varepsilon_r$ as a function of temperature at 100 Hz–1 MHz of the PZT films is examined and shown in Fig. 4g. It is observed that $\varepsilon_r$ are temperature independent at low temperature (<50 °C). Then, they all increase gradually with increasing temperature to their maximum value of around 245 °C, corresponding to the transition from a ferroelectric to a paraelectric phase. With the increase of measurement frequency, $\varepsilon_r$ takes on a firstly decreasing and then increasing change (Inset of Fig. 4g). Fig. S13 shows the polarization-electric field (P-E) ferroelectric hysteresis loops of the annealed PZT films at 30 °C and 10 Hz. The samples exhibit ferroelectricity with coercive field $E_C$ of ~10 KV cm$^{-1}$ and remnant polarization $P_r$ of ~30 μC cm$^{-2}$. The remanent polarization ($P_r$) of the EDP film is slightly higher than that of the dip-coated samples. We compare the piezo-electric coefficient $d_{33}$ of PZT films prepared by common methods with this work as a function of the accessible PZT film thickness (Fig. 4h and Table S6). EDP strategy and in-situ electrostatically crystallographic structure upgrading are capable of fabricating PZT films with the thickness up to 50 μm and effective piezoelectric coefficient of ~560 pm/V. A common technique to enhance piezoelectric properties

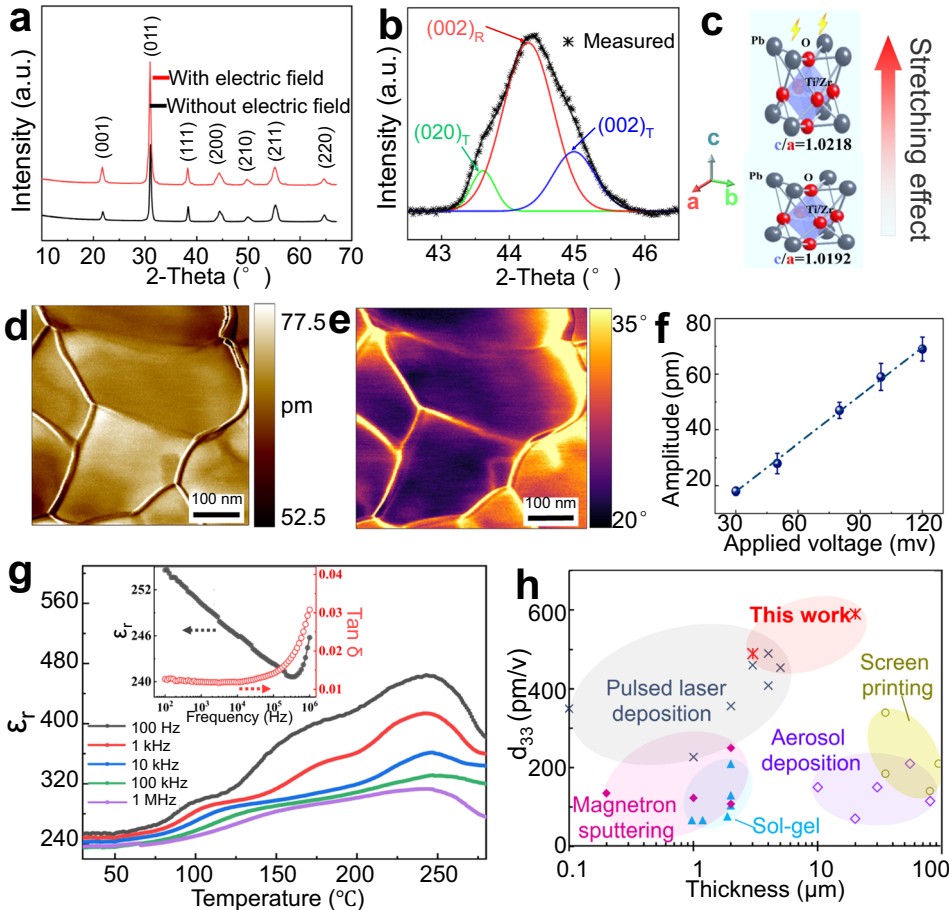

**Fig. 4 | Electrical characterizations of EDP deposited PZT films. a** X-ray diffraction patterns for PZT films fabricated by dip-coating and EDP. Additional results can be found in Fig. S12 and Table S5. **b** X-ray diffraction pattern from 42.5° to 46.5° and corresponding Gauss fitted patterns for PZT films fabricated by EDP. **c** Schematic of the increased tetragonal structure distortion of PZT induced by an electric field. **d** PFM amplitude map of the PZT films. **e** PFM phase map of the PZT films. **f** Linear dependence of PFM amplitude on the applied AC voltage. The slopes of the lines

provide effective piezoelectric coefficients of the annealed PZT films, which are approximately 590 pm V$^{-1}$. **g** Temperature dependence of relative dielectric constant $\varepsilon_r$ of the EDP deposited PZT films at different frequencies. Inset shows the frequency dependence of the relative dielectric constant $\varepsilon_r$ and dielectric loss tan δ for the PZT films at 30 °C. **h** Comparison of piezoelectric coefficient $d_{33}$ of PZT films prepared by existing techniques with this work as a function of the accessible PZT films thickness (<100 μm).

of films is to construct phase boundaries by tuning the composition chemically. However, the dopant content is only a few percent and often involves highly volatile elements (Pb, K, Na, and Bi), which hinders obtaining reliable and reproducible synthesis of films. Our strategy of electric field-induced structure distortion shows excellent potential for improving structural properties to get high-performance piezoelectric films.

## Piezoelectric devices fabricated via the EDP

A flexible piezoelectric sensor is manufactured using a cantilever vibration system (Fig. 5a, b). The mica substrate is adopted due to its mechanical flexibility, high thermal stability, chemical inertness, and compatibility with PZT films. The silver interdigital electrodes (IDEs) are used to harvest electric potential. Both the real-time recorded vertical displacement signal of the vibrating cantilever and the open-circuit voltage output of PZT film show a damping attenuation trend, and they have a coordinated variation (Fig. 5c). We further designed a PZT/P(VDF-TrEE) hierarchically interconnected piezocomposite textile (HIPCT) and used it as an energy harvester. PZT ink is firstly deposited onto the machine-knitted nylon textile to form PZT ceramic framework. The annealed framework is then coated by the P(VDF-TrFE) solution (Fig. 5d). Finally, copper meshes are attached to both sides of the piezoelectric composite as electrodes. The piezoelectric

composite shows high flexibility and can be easily bent around a pen (Fig. 5e). Its flexibility can be attributed to the hierarchical structures and the encapsulation of P(VDF-TrFE). The hierarchical structure is constituted by the submillimeter-scale intertwined multi-strand yarns, which are further formed by dozens of twisted fibers (Figs. 5f and S14). Fig. S15a presents the optical photograph of the HIPCT-based energy harvester, and the relation between its electrical output and the compressive pressure values/frequency is demonstrated in Fig. S15b–f. Under 2.2 MPa compressive pressure with the frequency of 30 Hz, the output voltage of the HIPCT increases steadily as the load resistance ranges from 1 kΩ to 100 MΩ. The maximum power density of ~200 μW cm⁻² is obtained at ~1 MΩ (Fig. 5g). The generated alternating current (AC) signals can be rectified to direct current (DC) signals for energy storage. The HIPCT-based energy harvester charged a 4.7 μF and 10 μF capacitor to 1.4 V and 0.7 V, respectively, within 60 s, excited by a compressive load of 1 MPa and 30 Hz (Fig. 5h). The high output voltage of the HIPCT allows 12 LEDs lighting up simultaneously through palm tapping (Fig. 5i and Supplementary Movie 2). The voltage response of the energy harvester keeps nearly unchanged in 28,000 compressing cycles, indicating its high durability (Fig. S15d). In addition, the output current of the HIPCT-based energy harvester as a function of the compressive pressure/resistance are measured, as shown in Fig. S15e, f. The output current increases gradually from

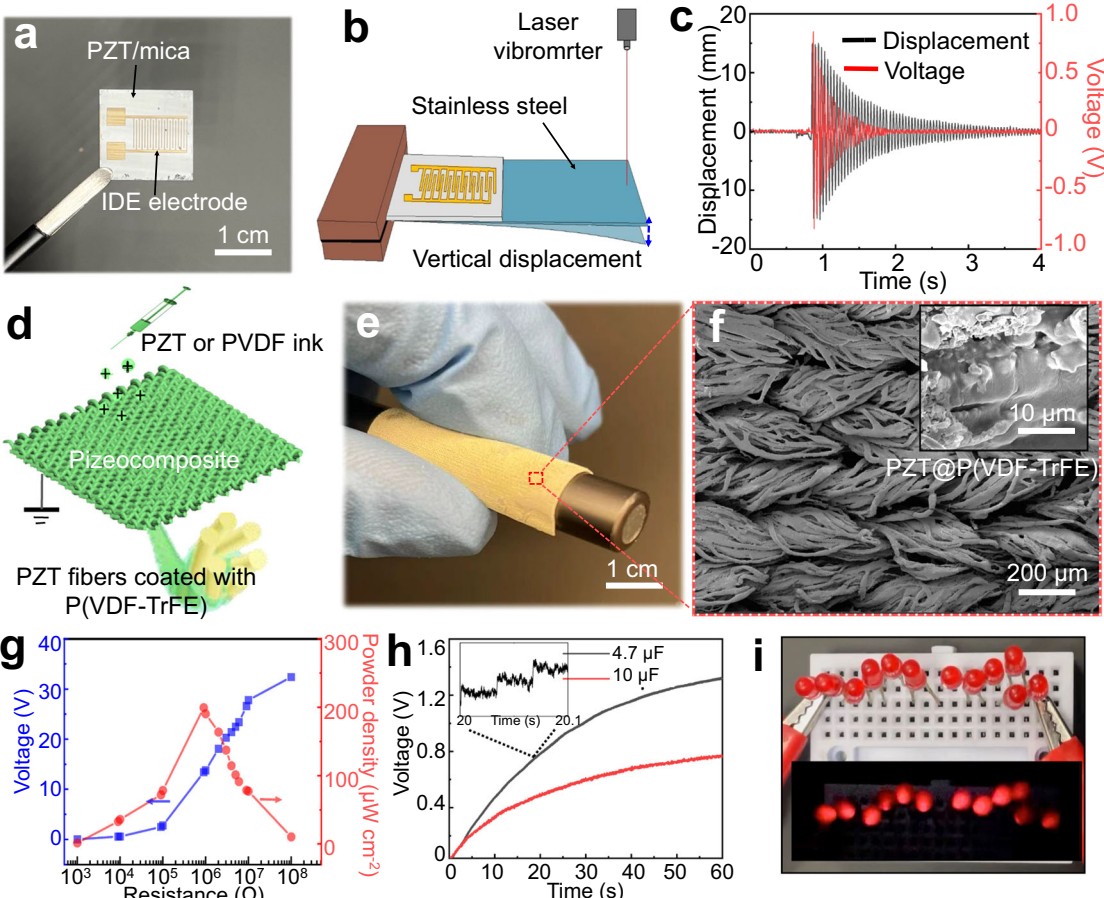

**Fig. 5 | Piezoelectric devices based on EDP PZT. a** Optical photograph of the EDP PZT film and IDE electrode. **b** Schematic of the piezoelectric sensor based on PZT films. **c** Voltage and vibration signals of the piezoelectric sensor based on a ~5 μm thick PZT films. Its open-circuit voltage output corresponds to the damped and attenuated vibration, and a maximum voltage of nearly 0.9 V. **d** Schematic of the fabrication of the piezo composite using a cloth-template assisted EDP. **e** An optical photograph of the piezo composite bent around a pen,

showing its high flexibility. **f** SEM images of piezo composite showing a hierarchical structure. **g** Output voltage and power density depend on the load impedance of 2.2 MPa compressive pressure with a frequency of 30 Hz. The maximum power output occurs at the resistance of ~1 MΩ. **h** The charging curve of the capacitors with 4.7 μF and 10 μF under the load impedance of 1 MPa compressive pressure with the frequency of 30 Hz. **i** Optical photograph of 12 LEDs lit up by the HIPCT device through palm tapping.

~15 μA to ~75 μA with the increase of the compressive pressure, while the output current decreases gradually from ~60 μA to ~1 μA with the increase of the resistance.

## Versatility and reproducibility of the EDP process

While most of the printed elements shown in the present work are PZT ceramics, other ceramics can be also printed by proper ink formulation. As a demonstration, we fabricate BTO (Barium Titanate) films (Fig. S16). Additionally, metals, polymers and biomaterials can also be fabricated by printing of inks containing metal nanoparticles, polymers or biomolecular precursors. As examples, Fig. S17–S19 show the silver, PVDF and glycine films. Printing these functional films of inks, ranging from suspensions of dielectric ceramic and metal nanoparticles, to insulating polymers, to solutions of biological molecules, illustrates the versatility of EDP process. Ultimately, the range of printable materials is only constrained by the requirement that the ink has proper physical properties. Therefore, by minor adjustments in inks properties, EDP can be extended to produce films, patterns, and particles from any materials, including biomaterials and even living cells.

As for all printing techniques, the process parameters should be optimized for the given ink and the reproducibility of the printed products is of principal importance. Here, the reproducibility and consistency of the EDP process is assessed by analyzing the piezoelectric and structural properties of PZT films/patterned lines that are printed via the optimized process parameters (Table S2). Three PZT films with the thickness of <10 μm are deposited at different times. PFM is used for piezoresponse measurements using an Asylum Cypher ES AFM system with a conductive probe (Nano world Arrow-EFM). For calculating piezoelectric coefficient $d_{33}$, an area (500 × 500 nm) was scanned in DART (dual AC resonance tracking) mode with varied tip drive voltage (from 10 mV to 200 mV), and the corresponding out-of-plane piezoelectric amplitudes are recorded over the scanned area. The maximum and minimum measured value is -470 and -648 pm V$^{-1}$, respectively. The mean value is -560 pmV$^{-1}$. Three samples with PZT linear structures arrays are fabricated via polyimide mask with ~100 μm-wide grooves. The widths of printed lines for each printed sample are measured at ten different positions. There is no statistically significant difference among any of the widths measured at the same sample (Fig. S21). These data indicate that our EDP can fabricate PZT films and micropatterns in a consistently uniform manner.

## Discussion

We present a versatile and fast microprinting technique by electrostatically triggering the liquid-air interface instability of inks through a conductive spiny disc. The EDP allows producing nanoparticles, micrometer-thick films, and micro-patterns of a broad range of materials. We demonstrate the printing speed up to $10^9$ μm$^3$ s$^{-1}$, surpassing all known fabricating techniques capable of depositing micrometer-thick films. Furthermore, this method can in-situ, electrostatically tune the crystallographic structures and gives high electrical and mechanical properties of the printed elements. In short, the EDP represents a significant step toward fast and large-area microprinting of diversified functional materials in many compositions and microstructures.

## Methods

### Ink formulation

Inks of PZT sol are prepared by the following steps: dissolving 7.39 g Lead (II) acetate trihydrate (Alfa Aesar, 99%) in the acetic acid (~6 g, Travel Circle International Co., Ltd., 99%); adding 3.45 g zirconium (IV) propoxide (Travel Circle International Co., Ltd., 70 wt.% in 1-propanol) and 2.45 g titanium (IV) butoxide (Travel Circle International Co., Ltd., 99%) into the solution under continuously stirring; 0.5 g ethylene glycol (Dieckmann, 99.5%) are added to adjust the surface tension. The

obtained inks have a molar ratio of 1.2 Pb: 0.52 Zr: 0.48 Ti. Inks of PZT slurry are prepared by adding the proper amount of PZT particles (typically 30–50 wt.%, Suzhou PANT Technology Co., Ltd.) in the PZT sol and homogenizing the mixture by high-energy ball milling. For the fabrication of PZT columnar structures, the PZT slurry concentration is 60–70 wt.%. Inks of BTO slurry are prepared by adding the proper amount of BTO particles (typically 30 wt.%, Suzhou PANT Technology Co., Ltd.) in the BTO sol and homogenizing the mixture by high-energy ball milling. For BTO sol, 2.55 g barium acetate (Travel Circle International Co., Ltd.) and 3.4 g tetrabutyl titanate (Travel Circle International Co., Ltd.) are mixed and dissolved in 6 ml ethylene glycol methyl ether. The solution is diluted to 0.5 M by 1 mL ethylene glycol (Alfa Aesar) and 1 mL distilled water. Inks of silver slurry are prepared by adding the proper amount of silver particles (typically 20 wt.%, Goodfellow) in the anhydrous ethanol and homogenizing the mixture by magnetic stirring. Inks of PVDF and P(VDF-TrFE) are prepared by dissolving the proper amount of Polyvinylidene fluoride (Sigma-Aldrich) or Poly(vinylidene-fluoride-cotrifluoroethylene) (VDF/TrFE = 70/30, Sigma-Aldrich) (typically 7–10 wt.%) in the dimethylformamide solution and homogenizing the mixture by magnetic stirring. Inks of glycine are prepared by dissolving the proper amount of glycine (typically 10 wt.%, Sigma-Aldrich) in the deionized water and homogenizing the mixture by magnetic stirring.

### Printer set-up and depositing process

Inks are loaded into a 2 ml syringe and supplied to the spiny disc through a drainage needle. The drainage needle is coated with an insulating film. The flow rate of the ink is controlled by a syringe pump (Cole-Parmer, Pump 74900-25). Ozone plasma surface treatment is applied to the drainage needle and spiny disc to improve liquid wetting. The spiny disc is obtained by laser cutting stainless steel foil. A high voltage supply (Dongwen, DW-P303-1ACH2) is used to apply a positive potential. At the same time, the substrate is positioned on an electrically grounded plate, mounted on an X-Y electric translation stage. The substrate is moved relative to the disc by the X-Y translation stage. A high-speed camera consisting of a 300X lens with adjustable zoom is used to monitor the jets. Jet stability is critical for printing reproducible products. It relies on the process parameters such as the applied voltage, ink properties, and environmental parameters, which, together with the flow rate, determine the size of the atomized droplets and the thickness of the printed layer. The films are built by alternately depositing of ink in the X and Y directions. To form highly dense films, we set the distance between the neighboring parallel paths of substrate to 4 mm to ensure partial overlap between deposited materials (Fig. S20). During the deposition of films and patterns, it is inevitable to enlarge the range of ejected droplets to undesired region for the uniformity of the films on the desired substrates. Given the fact that the deposited inks remain wet for a few seconds after printing, the consequent diffusion occurs among the adjacent composition (sol and powders), which is beneficial for the formation of the compact and poreless films. Sol-infiltration on every five layers of the as-deposited films by means of spin-coating (1000 rpm for 30 s) is performed. The films are then dried at 200 °C for 2 min to remove the organic components and reduce internal stress. The micropatterns are fabricated by dielectric mask-assisted EDP. The substrates are covered with polyimide films with a thickness of 200 μm, which are then patterned by an electronic craft cutting/engraving tool (Silhouette Cameo 4). During evaporation of the solvent in the deposited wet films, cracks appeared for the films deposited on the polyimide surface due to the sol induced shrinkage stresses. The mask can be easily removed after deposition without affecting the functionality of patterns. We calculated the coverage ratio of PZT on the target region and mask region by measuring the covered area of PZT via ImageJ. Printing is carried out under ambient conditions, with a relative humidity of 40% and temperature of ~25 °C. Sintering treatment at 800 °C for 30 min is

performed for synthesizing perovskite structures. Polarization of PZT films is conducted at 130 °C under 10 kV mm$^{-1}$ for 30 min. During the deposition PZT particles, the Si substrate is positioned on a heating plate with a surface temperature of 600 °C, and the disc-to-substrate distance greater than 40 mm.

## Material characterizations

The microstructures of samples are observed by scanning electron microscopy (SEM; FEI Quanta 450). The phase structures of the PZT films are measured by XRD (Rigaku SmartLab) with a scan speed of 2° min$^{-1}$. The ferroelectric hysteresis loops are obtained by a ferroelectric analyzer (PK-CPE 1701, PolyK Technologies, USA). The relative dielectric constant and corresponding dielectric loss tanδ are detected by a dielectric property test system (DPTS-AT-600, Wuhan Yanhe Technology Co., Ltd). PFM is used for piezoresponse measurements using an Asylum Cypher ES AFM system with a conductive probe (Nano world Arrow-EFM). For calculating effective piezoelectric coefficient $d_{33}$, a small area (500 × 500 nm) was scanned in DART (dual AC resonance tracking) mode. Drive voltage amplitude on the tip varies from 30 mV to 120 mV, and the corresponding out-of-plane piezoelectric signals are recorded and averaged over the scanned area. The leakage current behavior of our PZT films is measured under a DC bias field of 150 kV/cm and a temperatures of 180 °C (Supplementary Note S3 and Supplementary Fig. S22).

## Fabrication and characterization of sensor and energy harvester

PZT films are deposited on the 20 μm- thick mica substrates (Taiyuan Fluorophlogopite Mica Company Ltd.). Then silver IDEs with a width of 0.8 mm, finger spacing of 0.3 mm, finger length of 5 mm, and thickness of ~0.4 μm are printed on the surface of PZT films by mask-assisted EDP. The PZT film/mica, IDEs, and their connected wires are encapsulated by a thin layer of polydimethylsiloxane (PDMS) and glued to the end of a steel cantilever (90 mm × 20 mm × 0.3 mm). Polarization of PZT films is conducted at 130 °C under 10 kV mm$^{-1}$ for 30 min. The open-circuit output voltages are recorded by an oscilloscope (Rohde & Schwarz RTE1024). The damped attenuated vibration displacements are recorded by a laser vibrometer (Polytec NLV-2500). To fabricate the HIPCT device, the PZT framework is firstly obtained by printing PZT slurry onto the machine-knitted nylon textile and sintering at 1000 °C for 2 h. The P(VDF-TrFE) film is then deposited onto the annealed PZT framework. Afterwards, copper meshes (200 mesh) are attached to the top and bottom surfaces of the composite via hot pressing (~1 KPa) at 120 °C for 1 h. Finally, the assembled device is electrically poled with the electric field of 20 kV mm$^{-1}$ at 130 °C for 30 min. The compressing test was conducted by a shaker which compressing force can be detected and quantified by a mechanical force sensor. The output voltages are recorded by an oscilloscope (Rohde & Schwarz RTE1024).

## Finite Element Simulations

The simulation of the electric potential, electric field, fluid volume fraction, fluid velocity, and fluid pressure around the spiny disc (with 5 tips) during EDP (Figs. 1f and S1) is done in COMSOL using the following parameters: applied voltage of 4 kV; disc-to-substrate separation of 10 mm; disc diameter of 5 mm. The properties of inks used in this simulation are listed in Table S1.

## Data availability

The data that support the findings of this study have been included in the main text and Supplementary Information. All other relevant data supporting the findings of this study are available from the corresponding authors upon request. Source data are provided with this paper.

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

## Acknowledgements

This work was supported by General Research Grant (Project Nos. 11212021, 11210822 (Z.Y.)) from the Research Grants Council of the Hong Kong Special Administrative Region, and the Innovation and Technology Fund (ITS/065/20; GHP/096/19SZ (Z.Y.)) from the Innovation and Technology Commission of the Hong Kong Special Administrative Region.

## Author contributions

Conceptualization: X.L., Z.Z., Z.Y. Methodology: X.L., Z.Z., Z.Y. Investigation: X.L., Z.Z., Z.P., X.Y., Y.H., S.L., W.L., Y.S., Z.Y., Y.W. Visualization: X.L. Supervision: Z.Y. Writing—original draft: X.L. Writing—review & editing: Z.Z., Z.P., X.Y., Y.H., S.L., W.L., Y.S., Z.Y.

## Competing interests

The authors declare no competing interests.
