## [Peer Review File · Nature Communications]

Ultrafast and versatile electrostatic disc microprinting for piezoelectric elementsREVIEWER COMMENTS

Reviewer #1 (Remarks to the Author):

The authors present here electrostatic disc microprinting (EDP), a scalable microprinting strategy capable of producing a variety of piezoelectric particles, films and patterns. The electrostatic forces used by EDP in an air-liquid interface allow for deposition speeds of to $109 \mu\text{m}^3/\text{s}^{-1}$, which is one order of magnitude faster than comparable techniques. Additionally, the electric field applied during EDP leads to a field-induced lattice deformation that allows fabricated PZT films to exhibit piezoelectric performances greater than other comparable fabrication methods. Using masks, in a photolithographic fashion, the authors also created piezoelectric film patterns with microscale resolution. Finally, using a judicious and fine control of the fabrication parameters of the EDP process, the authors demonstrated the creation of nanoparticles with good monodispersity. The work presented here is not only original, but truly scalable, having the potential to transform current approaches to fabricate piezoelectric materials for wearable devices and IoT systems. The manuscript is nicely written and illustrated and the claims are mostly supported by the results. However, there are a few experimental details that remain obscure to the reader and I recommend the authors to address the following comments:

- 1- Line 63-64: "Such electrostatically driven cone-jetting phenomena occur widely in nature and application." This sentence is too vague (no application is mentioned until later) and reads weird. Please merge this sentence with the following one and re-write both, so that its split reads well.
- 2- Video 2: Please add some other labels indicating how the piezoelectric material is compressed. The video, otherwise, does not provide much information.
- 3- Figure 1a: This schematic should explain that the voltage applied to the tip is a positive voltage. It is, therefore, confusing why there are negative charges on the tip. The average tip-substrate distance used during the experiments should be mentioned.
- 4- The design of the "spiny disc" is not properly explained across the manuscript. What is the maximum radius of this disc? Is there any limitation in terms of the number of outlets? Would it be beneficial for this disc to spin? note that the authors called "spiny" but it doesn't spin! Please discuss all these issues in the manuscript, as they are critical to fully understand the fabrication method.
- 5- It is unclear if the piezoelectric material can be easily transferred (without losing its outstanding piezoelectric performance) from a conductive layer (necessary for its fabrication) to another substrate (non-conductive, for example). This should be discussed in the manuscript.
- 6- (In line with the previous comment) It is also unclear how the mask used to make micropatterns can be removed without removing the micropatterns too. The authors should explicitly mention this effect and explain that, due to the adhesion of PZT to the substrate, the minimum feature size was (insert-here-figure-of-merit).
- 7- The authors should improve the description of the materials used. It is unclear if the PZT or PVDT were purchased or synthesized. More importantly, it is unclear what was the viscosity of the ink and the overall role of this viscosity on the voltage threshold and the quality of the deposited materials. These details are very important and should be explicitly discussed across the text.

Reviewer #2 (Remarks to the Author):

In this manuscript, the authors developed ultrafast and versatile electrostatic disk microprinting(EDP) for piezoelectric elements. The EDP process could fabricate lead zirconate titanate(PZT) free-standing particles, films, and micro-patterns at a high speed of $109 \mu\text{m}^3/\text{s}$. The fabricated PZT films showed a high piezoelectric strain constant of $590 \text{ pm}/\text{V}$, which is $1\sim 2$ times

higher than previous research. In general, the experiment is well designed, and the conclusion is logically supported by the experimental results. However, there are still some issues to be addressed. It needs revisions and improvements for acceptance.

Comments:

1. The authors proposed electrostatic disc microprinting (EDP) with the spiny disc. However, spiny disc contained 16 outlets at the end tip of nozzle, and thereby, entire spiny disc could be considered a type of multi-nozzle that has shape of disc. Therefore, a word 'nozzle-free' should be removed or replaced to another word. Also, some articles are recommended when introduced the electrostatic printing technique with multiple nozzle or nozzle-free to improve large-scale production: 10.1016/j.jiec.2015.06.033; 10.1021/acsaem.7b00227.
2. Simulation with the simplified nozzle with 5 outlets were shown in Figure 1f and Figure S1. However, two Figures seem to have 6 outlets. Besides, the number of outlets could affect the deposition tendency of ejected materials. Could simplification for electrical simulation with fewer outlets of spiny disc explain the deposition tendency of as-deposited products? Also, how about the conformality of PZT films or micro-patterns in large-scale production? The following articles are relevant to conformal fabrication, which are suggested to be considered: 10.1007/s42765-023-00279-3; 10.1002/advs.202106030.
3. Ejected materials are highly charged and moved along the induced electric field. However, depending on the deposition range, speed, and tip-to-collector distance, the range of ejected materials may enlarge with being deposited on the undesired region or substrate. Can the undesirable deposition of as-deposited films or patterns be adjusted with just multiple prints using a X-Y mechanical stage and PZT sol spin-coating process?
4. The proposed EDP process enables the fabrication of large-scale and fast production of piezoelectric devices. However, electrostatic deposition using dielectric masks generally enabled to fabricate micro-patterns, but also deposition of used materials occurred on the dielectric mask, where unused areas. How about the fabrication efficiency of the products in terms of the material usage? The following articles are relevant to your work, which are suggested to be considered: 10.1016/j.ceramint.2020.07.249, 10.1038/s41598-021-01043-6.
5. In the Abstract, the authors declared that the EDP process allows for one-step fabrication of PZT nanoparticles, films, and micro-patterns. However, spin-coating with PZT sol conducted to further improve the density of the as-deposited films. Without the spin-coating process, can PZT films and micro-patterns be fabricated well? If not, the word 'one-step' should be applied to fabricate only PZT nanoparticles in the whole manuscript or removed to avoid misunderstanding to readers.
6. In general, the performance of energy harvesters includes not only output voltage but also output current. However, in the paper, only the output voltage performance graph is showed. Therefore, to show the device performance, the current graph according to variable, such as frequency and force, is recommended to be added.
7. In general, the performance of a pressure sensor is verified by sensitivity, such as the amount of generated power (V/N or V/Pa) according to the pressure (force) and R-squared score. However, in Figure S11, which shows the performance as the pressure sensor, it is difficult to know quantitatively the sensitivity of the device. Correspondingly, it is recommended to add the above two indicators for the quantitative inspection.
8. The Figure 3e is not match the manuscript. Also, some typos and grammar error are confirmed in the manuscript and figures. The authors should be corrected throughout the text and figures.

Reviewer #3 (Remarks to the Author):

The work entitled „Ultrafast and Versatile Electrostatic Disc Microprinting for Piezoelectric Elements” submitted by Li, Zhang, et al. is very innovative and without doubt of high interest for the scientific community in the field of functional materials and related technologies. However, at least in my view, the paper shows some weaknesses in detail, that should be addressed by the authors prior to publication.

The reported work in the main deals with deposition technology, and in my role as a technologist I miss some concrete information about the presented approach compared to other work in the field.

One of the core features of the proposed technology, as far as I understand, is the self-polarizing

feature of the fabricated films and patterns. In that respect I miss the classification in comparison with other work in this field, e.g., the work of Khan et al. recently published in 2022 [Khan, Rukhsar Ali, et al. "Development of self-polarized PVDF films on carbon fabrics for sensing applications." *The Journal of The Textile Institute* 113.10 (2022): 2208-2214]. Besides, when discussing compatibility of common patterning techniques with flexible and stretchable substrates, the authors should mention PVDF and related co-polymers more explicitly in the state of the art, since these materials seem increasingly important in the relevant field.

On page 5 (line 209) the authors claim, that their technology facilitates the realization of "high-quality" piezoelectric films. However, the label "high-quality" of thin films comprises more than a high piezoelectric coefficient, it is also concerned with things such as surface roughness, achievable layer uniformity and homogeneity, and the constraints with regard to substrate dimensions that arise as a consequence.

Authors should also comment on issues like reproducibility (piezoelectric and structural properties) in fabrication, as well as on durability and aging effects of the fabricated PZT structures. For example, a well-known problem of PZT is resistance degradation under electrical field load, additionally strongly dependent on the used electrode material. How do the layers fabricated by the proposed technology perform in this respect?

Thank you for your review and constructive comments on the manuscript (NCOMMS-23-19123-T) entitled "*Ultrafast and Versatile Electrostatic Disc Microprinting for Piezoelectric Elements*" submitted for publication on Nature Communications. We have revised the manuscript carefully. All changes in the revised manuscript text file have been marked with red color. Please find responses to reviewers' comments below.

Reviewer Comments (and change made in accordance)

Reviewer #1 (Remarks to the Author):

The authors present here electrostatic disc microprinting (EDP), a scalable microprinting strategy capable of producing a variety of piezoelectric particles, films and patterns. The electrostatic forces used by EDP in an air-liquid interface allow for deposition speeds of to 10^9 $\mu\text{m}^3/\text{s}^{-1}$, which is one order of magnitude faster than comparable techniques. Additionally, the electric field applied during EDP leads to a field-induced lattice deformation that allows fabricated PZT films to exhibit piezoelectric performances greater than other comparable fabrication methods. Using masks, in a photolithographic fashion, the authors also created piezoelectric film patterns with microscale resolution. Finally, using a judicious and fine control of the fabrication parameters of the EDP process, the authors demonstrated the creation of nanoparticles with good monodispersity. The work presented here is not only original, but truly scalable, having the potential to transform current approaches to fabricate piezoelectric materials for wearable devices and IoT systems. The manuscript is nicely written and illustrated and the claims are mostly supported by the results. However, there are a few experimental details that remain obscure to the reader and I recommend the authors to address the following comments:

1- Line 63-64: "Such electrostatically driven cone-jetting phenomena occur widely in nature and application." This sentence is too vague (no application is mentioned until later) and reads weird. Please merge this sentence with the following one and re-write both, so that its split reads well.

Response:

We thank the referee for the careful review and professional comments. We have corrected the sentences in the revised manuscript on line 48-51, "Such electrostatically driven cone-jetting phenomena occur widely in nature and application, and two well-known examples are the ejection of streams of charged droplets from the tips of raindrops in a thunderstorm cloud and one immensely popular application for assaying large biomolecules: electrospray mass spectrometry."

2-Video 2: Please add some other labels indicating how the piezoelectric material is compressed. The video, otherwise, does not provide much information.

Response:

We thank the referee for the careful review and professional comments. 12 LEDs are lit up by our HIPCT device through palm tapping. We have corrected the Video S2 in the revised manuscript as suggested.

3-Figure 1a: This schematic should explain that the voltage applied to the tip is a positive voltage. It is, therefore, confusing why there are negative charges on the tip. The average tip-substrate distance used during the experiments should be mentioned.

Response:

We thank the referee for the careful review and professional comments. Figure 1a has been modified as suggested. During the EDP process, two types of charge transport (charge conduction and charge convection) occur when the fluid flows through the tip. As shown in Figure 1a, the electrons of the fluid move to the wall of DC (+) voltage-attached tip, and causes the electric normal stress ($F_{E, n}$). The surface charges induced by charge conduction lead to electric tangential stress ($F_{E, t}$) along the surface meniscus with acceleration toward the cone apex. The charge acceleration will speed up the surrounding fluid sequentially. Charge convection occurs when the fluid is subjected to voltage gradient and the flow of ions is generated. This ions flow can carry the surface charge along with it, leading to the charge transport. We have added the description in the revised manuscript on line 98-101, “The electrons of the fluid move to the wall of DC (+) voltage-attached tip, and causes the electric normal stress ($F_{E, n}$). The surface charges induced by charge conduction lead to electric tangential stress ($F_{E, t}$) along the surface meniscus with acceleration toward the cone apex (Fig. 1a).”

The average disc-substrate distance used for depositing PZT films/patterns and nanoparticles is ~5 mm and ~60 mm, respectively. We have added the description in the revised manuscript on line 111-112, “The average disc-substrate distance is ~5 mm, and the other optimized process parameters are listed in Table S2.” and line 199-200, “The optimized parameters for one-step depositing PZT nanoparticles are listed in Table S4.”

Fig. 1a Schematic of cone-jetting formation from one tip.

Table S2. Optimized process parameters for EDP depositing PZT films/patterns.

Parameter	Value
Ink concentration (mass ratio of particles to sol)	30-50 wt.%
Depositing speed (mm s^{-1})	5~30
Distance between disc to substrate (mm)	2~8
Supply rate of ink ($\mu\text{l min}^{-1}$)	3-200
Applied voltage (kV)	3.0~10.0

Table S4. Optimized process parameters for EDP depositing PZT nanoparticles.

Parameter	Value
Sol concentration (M)	0.1-0.4
Substrate temperature ($^{\circ}\text{C}$)	600
Distance between disc to substrate (mm)	40-80
Supply rate of ink ($\mu\text{l min}^{-1}$)	1-10
Applied voltage (kV)	6.0~10.0

4-The design of the "spiny disc" is not properly explained across the manuscript. What is the

maximum radius of this disc? Is there any limitation in terms of the number of outlets? Would it be beneficial for this disc to spin? Note that the authors called "spiny" but it doesn't spin! Please discuss all these issues in the manuscript, as they are critical to fully understand the fabrication method.

Response:

We thank the referee for the careful review and professional comments. The dimensions of the thin spiny disc used in our work is shown in Fig. S1a. Its diameter of addendum circle and dedendum circle is 5.5 mm and 5 mm, respectively. The topology of the multi-tips design helps trigger liquid-air interface instability at the rim of the disc, which is critical to generating multiple radial liquid ligaments.

By applying positive potential to different wetted spiny discs (spiny disc design in Fig.S1), multiple liquid jets are issued from the tips of disc, forming a symmetric radial jet mode (Fig. S2 and Fig. S3). As the increase of the disc diameter (D , from 3.5 mm to 10.5 mm), the applied voltage for generating stable cone-jets raise. For the small discs ($D=3.5$ mm), the ink is easy to flow out of the disc and drop onto the substrate, which inevitably will disrupt the uniformity of the deposited film. The liquid jet undergo Rayleigh-Plateau instability and will be split into droplet clusters with a diameter twice that of the jet. The amount of atomized droplets can be fine-tuned by controlling the number of tips (N). We find a well-defined optimum in atomization stability and productivity around $N=16$. At small N , atomizing yield is inhibited by the decreasing role of tip streaming; at high N , the mutual interference of jets/droplets is stronger, which affects the stability of jets.

In the rotation process of the disc, the centrifugal forces combined with the electric field force can elongate the liquid jet, which reduces the dependence on applied voltage and helps to increase the productivity. However, for our printing head, fluid stability on the disc is the primary issue to be addressed for the centrifugal assisted EDP.

We have added the description in the revised Supplementary Information (Supplementary Text Note S1 Formation of multiple jets) on line 60-73, “The dimensions of the thin spiny disc used in our work is shown in Fig. S1a. Its diameter of addendum circle and dedendum circle is 5.5 mm and 5 mm, respectively. The topology of the multi-tips design helps trigger liquid-air interface instability at the rim of the disc, which is critical to generating multiple radial liquid ligaments.

By applying positive potential to different wetted spiny discs (spiny disc design in Fig.S1), multiple liquid jets are issued from the tips of disc, forming a symmetric radial jet mode (Fig.

S2 and Fig. S3). As the increase of the diameter of disc (D , from 3.5 mm to 10.5 mm), the applied voltage for generating stable cone-jets raise. For the small discs ($D=3.5$ mm), the ink is easy to flow out of the disc and drop onto the substrate, which inevitably will disrupt the uniformity of the deposited film. The liquid jet undergo Rayleigh-Plateau instability and will be split into droplet clusters with a diameter twice that of the jet. The amount of atomized droplets can be fine-tuned by controlling the number of tips (N). We find a well-defined optimum in atomization stability and productivity around $N=16$. At small N , atomizing yield is inhibited by the decreasing role of tip streaming; at high N , the mutual interference of jets/droplets is stronger, which affects the stability of jets.”

Fig. S1. Dimensions of thin spiny discs. Disc with maximum radius of 5.5 mm and 16 tips (a), disc with maximum radius of 5.5 mm and 6 tips (b), disc with maximum radius of 5.5 mm and 30 tips (c), disc with maximum radius of 3.5 mm and 16 tips (d), and disc with maximum radius of 10.5 mm and 16 tips (e).

Fig. S2. Multiplexed EDP tip jetting from the thin disc with diameter (D) of 3.5 mm and tips number (N) of 16 (a). Droplets flowing out of the disc ($D=3.5$, $N=16$) (b). Multiplexed EDP tip jetting from the thin disc ($D=5.5$ mm, $N= 5$) (c). Multiplexed EDP tip jetting from the thin disc ($D=5.5$ mm, $N= 16$) (d). Multiplexed EDP tip jetting from the thin disc ($D=5.5$ mm, $N= 30$) (e). Multiplexed EDP tip jetting from the thin disc ($D=10.5$ mm, $N= 16$) (f). The applied voltage for generating stable cone-jets raise as the increase of the disc diameter. Liquid: ethanol; supply rate of ink: $10 \mu\text{l min}^{-1}$; distance between disc and substrate: ~ 5 mm.

5- It is unclear if the piezoelectric material can be easily transferred (without losing its outstanding piezoelectric performance) from a conductive layer (necessary for its fabrication) to another substrate (non-conductive, for example). This should be discussed in the manuscript.

Response:

We thank the referee for the careful review and professional comments. The piezoelectric films and patterns are directly deposited onto the alumina substrates and polyethylene terephthalate (PET) substrates (or other non-conductive substrates). The sediment thickness (within $100 \mu\text{m}$) and area (within 5×5 cm) are far smaller than the depositing distance (> 2 mm) and the area of electrically-ground stage (with a diameter of 20 cm), respectively. Therefore, the substrates and dielectric masks have very limited effect on the electric field between disc and ground electrode. The stable atomizing state is maintained throughout the entire deposition process. The PZT patterns deposited on the PET substrates shown in this work (Fig.3c, e and Fig. S9) are only dried at $200 \text{ }^\circ\text{C}$ for 2 min. EDP-derived piezoelectric films will be crystallized at low temperature by laser assisted annealing in our future work. We have added the description in the revised manuscript on line 178-183, “The patterns are directly deposited onto the polyethylene terephthalate (PET) substrates. The sediment thickness (within $100 \mu\text{m}$) and area (within 5×5 cm) are far smaller than the depositing distance (> 2 mm) and the area of

electrically-ground stage (with a diameter of 20 cm), respectively. Therefore, the substrates and dielectric masks have very limited effect on the electric field between disc and ground electrode. The stable atomizing state is maintained throughout the entire deposition process. The PZT patterns deposited on the PET substrates are only dried at 200 °C for 2 min.”

6- (In line with the previous comment). It is also unclear how the mask used to make micropatterns can be removed without removing the micropatterns too. The authors should explicitly mention this effect and explain that, due to the adhesion of PZT to the substrate, the minimum feature size was (insert-here-figure-of-merit).

Response:

We thank the referee for the careful review and professional comments. The micropatterns are fabricated by dielectric mask-assisted EDP. The substrates are covered with polyimide films with a thickness of 200 μm , which are then patterned by an electronic craft cutting/engraving tool (Silhouette Cameo 4). By keeping the substrate electrically-ground well, we electrostatically guide the nanodroplets towards the attractive area (holes of the mask), whilst being away from the repulsive area (dielectric film) (Fig. 3a). Because the attractive area is recessed relative to the repulsive area, charged particles accumulation around the edge of the holes (attractive areas) occurs and will prevent nanoparticles deposition near the edge of the attractive area. This funneling effect will reduce the width of the area coated with deposited nanoparticles, which can push the resolution of assembly beyond that of the defined dielectric mask. During evaporation of the solvent in the wet films, cracks appeared for the films deposited on the polyimide surface due to the sol induced shrinkage stresses. The mask can be easily removed after deposition without affecting the functionality of the patterns. (Fig. S10a). The sloping side walls for deposited patterns are formed due to the funneling effect (Fig. S10b, c). Previously such features were attributed to edge fractures during masks removal. However, after examination of the similar cross sections with the masks, it can be seen that these sloping side walls still exist and are caused due to the funneling effect (particles located at the edge of the mask shielding the subsequent particles deposition) and the diffusion of wet films. Given the fact that the deposited inks remain wet for a few seconds after printing, the consequent diffusion of wet films occurs. We have added the description in the revised manuscript on line 152-161, “Charged nanoparticles and droplets, generated by EDP, are simultaneously deposited onto the substrates, which are covered by the dielectric mask (Fig. S8). By keeping the substrate electrically-ground well, we electrostatically guide the nanoparticles towards the attractive area (holes of the dielectric mask or target region), whilst being away from the

repulsive area (dielectric film or mask region) (Fig. 3a). Because the attractive area is recessed relative to the repulsive area, charged particles accumulation around the edge of each hole (mask region) occurs and will prevent nanoparticles deposition near the edge of the attractive area. This funneling effect will reduce the width of the area coated with deposited nanoparticles, which can push the resolution of assembly beyond that of the defined dielectric mask. The mask can be easily removed with tweezers following deposition and heat treatment without affecting the functionality of the patterns.” and line 167-168, “The sloping side walls for deposited patterns are formed due to the funneling effect and the diffusion of wet films (Fig. S10).”

Fig. S10. The processing steps to fabricate micropatterns (a). Line profile taken from the AFM image of the PZT surface shown in Fig.3d (b). Line profile taken from the AFM image of the Ag surface shown in Fig.3d (c).

7- The authors should improve the description of the materials used. It is unclear if the PZT or PVDF were purchased or synthesized. More importantly, it is unclear what was the viscosity of the ink and the overall role of this viscosity on the voltage threshold and the quality of the deposited materials. These details are very important and should be explicitly discussed across

the text.

Response:

We thank the referee for the careful review and professional comments. Inks of PZT slurry are prepared by adding the proper amount of commercial PZT nanoparticles (typically 30-50 wt.%, Suzhou PANT Technology Co., Ltd.) in the self-made PZT sol and homogenizing the mixture by 20 hours high-energy ball milling. The PZT sol are prepared by the following steps: dissolving Lead (II) acetate trihydrate (Alfa Aesar, 99%) in the acetic acid (TCI, 99%); adding zirconium (IV) propoxide (TCI, 70 wt.% in 1-propanol) and titanium (IV) butoxide (TCI, 99%) into the solution under continuously stirring; some amounts of ethylene glycol (Dieckmann, 99.5%) are added to adjust the surface tension. The obtained inks have a molar ratio of 1.2Pb: 0.52Zr: 0.48Ti. Inks of PVDF and P(VDF-TrFE) are prepared by dissolving the proper amount of Polyvinylidene fluoride (Sigma-Aldrich) or Poly(vinylidene fluoride-co-trifluoroethylene) (VDF/TrFE=70/30, Sigma-Aldrich) (typically 7-10 wt.%) in the dimethylformamide solution and homogenizing the mixture by magnetic stirring. We have added these descriptions in Materials and Methods section of the revised manuscript on line 298-304, “Inks of PZT sol are prepared by the following steps: dissolving Lead (II) acetate trihydrate (Alfa Aesar, 99%) in the acetic acid (TCI, 99%); adding zirconium (IV) propoxide (TCI, 70 wt.% in 1-propanol) and titanium (IV) butoxide (TCI, 99%) into the solution under continuously stirring; some amounts of ethylene glycol (Dieckmann, 99.5%) are added to adjust the surface tension. The obtained inks have a molar ratio of 1.2Pb: 0.52Zr: 0.48Ti. Inks of PZT slurry are prepared by adding the proper amount of PZT particles (typically 30-50 wt.%, Suzhou PANT Technology Co., Ltd.) in the PZT sol and homogenizing the mixture by high-energy ball milling.” and line 309-312, “Inks of PVDF and P(VDF-TrFE) are prepared by dissolving the proper amount of Polyvinylidene fluoride (Sigma-Aldrich) or Poly(vinylidene fluoride-co-trifluoroethylene) (VDF/TrFE=70/30, Sigma-Aldrich) (typically 7-10 wt.%) in the dimethylformamide solution and homogenizing the mixture by magnetic stirring.”

Some physical properties (including viscosity, surface tension, electrical conductivity, and relative permittivity and) of the PZT inks are measured and listed in Table S1. The viscosity of the PZT slurry is measured using an Ubbelohde viscometer. The surface tension is measured by a contact angle meter (Krüss DSA 100, Krüss GMBH). The relative permittivity is obtained by a precision impedance analyser (4294A, Agilent Technologies). The electrical conductivity is measured by a conductive meter (DDS-307, Shanghai INESA Scientific Instrument). The material properties suitable for obtaining a stable cone-jet mode used in electrohydrodynamic tip streaming process require an electrical conductivity of more than 10^{-11} S m^{-1} , a surface

tension of less than 50 mN m^{-1} and a viscosity of less than 100 mPa s^{-1} [4]. It can be seen from Table S1 that our PZT inks meet the requirements for obtaining a stable cone-jet mode. We have added these descriptions in Table S1 of the revised supplementary information on line 345-354.

The collision and dispersion of droplets on the substrate play an important role in the evolution of the as-deposited structures. For the high-concentration inks, atomized solvent-depleted particles tend to form a ‘powdery’ deposit and be exceedingly agglomerated owing to their poor mobility. These resident particle clusters will attract subsequent particles to agglomerate to them under the electrical field, referring to preferential landing. For the low-concentration inks, atomized particles enveloped by sol can improve the flow activity of the deposited wet films, which will increase the bonding behavior of the particles and reduce the porosity of films. Fig. S4 shows the surface characteristics of the deposited PZT films using ink of different concentrations. The films deposited using the 70 wt.% PZT slurry show porous feature. With the decrease of slurry concentrations (50-30 wt.%), the porosity of films decreases. When the slurry concentration is further decreased, the films show distinct cracks due to the high level of shrinkage induced stress in the sol during drying and pyrolysis. The viscosity of PZT slurry with different concentrations (70 wt.%, 50 wt.%, 30 wt.%, and 10 wt.%) is ~ 1.6 , ~ 1.4 , ~ 1.1 , and $\sim 1.0 \text{ mPa s}$. The applied voltages for stable EDP process using different inks (concentrations of 70 wt.%, 50 wt.%, 30 wt.%, and 10 wt.%) are ~ 3 to $\sim 10 \text{ kV}$. We have added these descriptions in Fig. S4 of the revised supplementary information on line 153-161.

Fig. S4. SEM images of surface topography of the deposited PZT films using PZT ink with different concentrations: 70 wt.% (a); 50 wt.% (b); 30 wt.% (a); 10 wt.% (a). The films deposited using the 70 wt.% PZT slurry show porous feature. With the decrease of slurry concentrations (50-30 wt.%), the porosity of films decreases. When the slurry concentration is further decreased, the films show distinct cracks due to the high level of shrinkage induced stress in the sol during drying and pyrolysis. The viscosity of PZT slurry with different concentrations (70 wt.%, 50 wt.%, 30 wt.%, and 10 wt.%) is ~1.6, ~1.4, ~1.1, and ~1.0 mPa s. The applied voltages for stable EDP process using different inks (concentrations of 70 wt.%, 50 wt.%, 30 wt.%, and 10 wt.%) are ~3 to ~10 kV.

Table S1. Physical properties of PZT inks (50 wt.%) for films and patterns fabrication. Experimental physical properties of PZT inks are measured at 25 °C.

Viscosity (10^{-3} Pa s)	Surface tension (10^{-3} N m $^{-1}$)	Relative permittivity	Electrical conductivity (10^{-3} S m $^{-1}$)
1.40	22.9	9.5	6.5

[4] Smith, D. P. H. The Electrohydrodynamic Atomization of Liquids. IEEE Trans. Ind. Appl. IA-22, 527–535 (1986).

Reviewer #2 (Remarks to the Author):

In this manuscript, the authors developed ultrafast and versatile electrostatic disk microprinting (EDP) for piezoelectric elements. The EDP process could fabricate lead zirconate titanate (PZT) free-standing particles, films, and micro-patterns at a high speed of $10^9 \mu\text{m}^3/\text{s}$. The fabricated PZT films showed a high piezoelectric strain constant of 590 pm/V, which is 1~2 times higher than previous research. In general, the experiment is well designed, and the conclusion is logically supported by the experimental results. However, there are still some issues to be addressed. It needs revisions and improvements for acceptance.

Comments:

1. The authors proposed electrostatic disc microprinting (EDP) with the spiny disc. However, spiny disc contained 16 outlets at the end tip of nozzle, and thereby, entire spiny disc could be considered a type of multi-nozzle that has shape of disc. Therefore, a word 'nozzle-free' should be removed or replaced to another word. Also, some articles are recommended when introduced the electrostatic printing technique with multiple nozzle or nozzle-free to improve large-scale production: 10.1016/j.jiec.2015.06.033; 10.1021/acsaem.7b00227.

Response:

We thank the referee for the careful review and professional comments. The word 'nozzle-free' has been removed in the revised manuscript. We have added the description of electrostatic printing technique with multiple nozzle or nozzle-free in the Introduction section of the revised manuscript on line 56-61, "For the nozzle-based printing strategies, ink flow defined by the inner diameters of nozzles, as well as the capillary phenomena put an upper limit on the printing speed. Additionally, nozzles with smaller apertures suffer from clogging and viscous losses, consequently, inks are limited to low-viscosity solutions free of large particles, which limits material versatility. To unleash the potential of high-speed and versatile printing allowed by electrostatic printing, some strategies based on multiple nozzle or nozzle-free have been proposed^{32,33}."

32. Shi, K. & Giapis, K. P. Scalable Fabrication of Supercapacitors by Nozzle-Free Electrospinning. ACS Appl. Energy Mater. 1, 296–300 (2018).

33. Kim, I. G., Lee, J.-H., Unnithan, A. R., Park, C.-H. & Kim, C. S. A comprehensive electric field analysis of cylinder-type multi-nozzle electrospinning system for mass production of nanofibers. J. Ind. Eng. Chem. 31, 251–256 (2015).

2. Simulation with the simplified nozzle with 5 outlets were shown in Figure 1f and Figure S1. However, two Figures seem to have 6 outlets. Besides, the number of outlets could affect the deposition tendency of ejected materials. Could simplification for electrical simulation with fewer outlets of spiny disc explain the deposition tendency of as-deposited products? Also, how about the conformity of PZT films or micro-patterns in large-scale production? The following articles are relevant to conformal fabrication, which are suggested to be considered: 10.1007/s42765-023-00279-3; 10.1002/advs.202106030.

Response:

We thank the referee for the careful review and professional comments. “five tips” has been corrected to “six tips”.

By applying positive potential to different wetted spiny discs (spiny disc design in Fig.S1), multiple liquid jets are issued from the tips of disc, forming a symmetric radial jet mode (Fig. S2). As the increase of the disc diameter (D , from 3.5 mm to 10.5 mm), the applied voltage for generating stable cone-jets raise. For the small discs ($D=3.5$ mm), the ink is easy to flow out of the disc and drop onto the substrate, which inevitably will disrupt the uniformity of the deposited film. The liquid jet undergo Rayleigh-Plateau instability and will be split into droplet clusters with a diameter twice that of the jet. The amount of atomized droplets can be fine-tuned by controlling the number of tips (N). We find a well-defined optimum in atomization stability and productivity around $N=16$. At small N , atomizing yield is inhibited by the decreasing role of tip streaming; at high N , the mutual interference of jets/droplets is stronger, which affects the stability of jets. The simplification of electrical simulation with six tips of the spiny disc can avoid mutual interference of jets, even though it causes a decrease in droplets depositing ratio. We have added the description in the revised Supplementary Information (Supplementary Text Note S1 Formation of multiple jets) on line 60-73, “The dimensions of the thin spiny disc used in our work is shown in Fig. S1a. Its diameter of addendum circle and dedendum circle is 5.5 mm and 5 mm, respectively. The topology of the multi-tips design helps trigger liquid-air interface instability at the rim of the disc, which is critical to generating multiple radial liquid ligaments.

By applying positive potential to different wetted spiny discs (spiny disc design in Fig.S1), multiple liquid jets are issued from the tips of disc, forming a symmetric radial jet mode (Fig. S2 and Fig. S3). As the increase of the diameter of disc (D , from 3.5 mm to 10.5 mm), the applied voltage for generating stable cone-jets raise. For the small discs ($D=3.5$ mm), the ink is easy to flow out of the disc and drop onto the substrate, which inevitably will disrupt the uniformity of the deposited film. The liquid jet undergo Rayleigh-Plateau instability and will

be split into droplet clusters with a diameter twice that of the jet. The amount of atomized droplets can be fine-tuned by controlling the number of tips (N). We find a well-defined optimum in atomization stability and productivity around N=16. At small N, atomizing yield is inhibited by the decreasing role of tip streaming; at high N, the mutual interference of jets/droplets is stronger, which affects the stability of jets. ”

The demand for 3D conformal fabrication is increasingly growing, driven by the potential of integrating functional materials on a curvilinear surface for fundamentally new characteristic. As one solution chemistry design-derived deposition, our EDP process has strengths in massive production. For large-area thin films or micropatterns, EDP is more suitable than spin or dip coating since there are no limitations on the size or geometry of the substrate. Note that 3D substrates with complex geometries should preferably be electroconductive for massive production. Given the fact that the deposited inks remain wet for a few seconds after printing, the consequent diffusion of wet films occurs. So the deposited material is easily to spread to the recessed region of the substrates and achieve completely covering of the 3D electroconductive surface. Here, we use EDP to conformally and uniformly deposit PZT films and patterns on the 3D free-form stainless steel substrates, cloth substrates and wrinkled steel sheet (Fig. 3e and Fig. S7). High-precision conformal printing of films and patterns should using a moving stage with the capability of robotic control of substrates and complex printing algorithms. We have added the description in the revised manuscript on line 120-129, “The demand for 3D conformal fabrication is increasingly growing, driven by the potential of integrating functional materials on a curvilinear surface for fundamentally new characteristic^{37,38}. Considering that the height step of each deposited layer ($< 4 \mu\text{m}$) is relatively small compared to the depositing distance, the effect of the reverse charging from as-deposited products to electric field is too small to disturb the precise layer-by-layer assembly. The deposited inks remain wet for a few seconds after printing, so the sediment will tend to spread to the recessed region of the substrates, which contribute to achieve 3D conformal printing. We use EDP to conformally and uniformly deposit PZT films on the 3D free-form stainless steel substrates, cloth substrates and wrinkled steel sheets (Fig. S7), demonstrating promising potential in 3D conformal electronics and smart textiles.”

Fig. S1. Dimensions of thin spiny discs. Disc with maximum radius of 5.5 mm and 16 tips (a), disc with maximum radius of 5.5 mm and 6 tips (b), disc with maximum radius of 5.5 mm and 30 tips (c), disc with maximum radius of 3.5 mm and 16 tips (d), and disc with maximum radius of 10.5 mm and 16 tips (e).

Fig. S2. Multiplexed EDP tip jetting from the thin disc with diameter (D) of 3.5 mm and tips number (N) of 16 (a). Droplets flowing out of the disc (D=3.5, N=16) (b). Multiplexed EDP tip jetting from the thin disc (D=5.5 mm, N= 5) (c). Multiplexed EDP tip jetting from the thin disc (D=5.5 mm, N= 16) (d). Multiplexed EDP tip jetting from the thin disc (D=5.5 mm, N= 30) (e). Multiplexed EDP tip jetting from the thin disc (D=10.5 mm, N= 16) (f). The applied voltage for generating stable cone-jets raise as the increase of the disc diameter. Liquid: ethanol; supply rate of ink: $10 \mu\text{l min}^{-1}$; distance between disc and substrate: $\sim 5 \text{ mm}$.

Fig. S7. Optical photographs of PZT films deposited on spherical surface (a) and the corresponding SEM image (b). Optical photograph of PZT films deposited on cloth substrate (c) and the corresponding SEM images (d). Optical photograph of PZT films deposited on wrinkled steel sheet (e) and the corresponding SEM images (f).

34. Liu, S. *et al.* 3D Conformal Fabrication of Piezoceramic Films. *Adv. Sci.* **9**, 2106030 (2022).
35. Song, J. Y., Kim, S., Park, J. & Park, S. M. Highly Efficient, Dual-Functional Self-Assembled Electrospun Nanofiber Filters for Simultaneous PM Removal and On-Site Eye-Readable Formaldehyde Sensing. *Adv. Fiber Mater.* **5**, 1088–1103 (2023).

3. Ejected materials are highly charged and moved along the induced electric field. However, depending on the deposition range, speed, and tip-to-collector distance, the range of ejected materials may enlarge with being deposited on the undesired region or substrate. Can the

undesirable deposition of as-deposited films or patterns be adjusted with just multiple prints using an X-Y mechanical stage and PZT sol spin-coating process?

Response:

We thank the referee for the careful review and professional comments. During the deposition of films and patterns, it is inevitable to enlarge the range of ejected droplets to undesired region for the uniformity of the films on the desired substrates. For example, to obtain a uniform film on a 20×20 mm substrate, the range of the ejected droplets (the moving range of the X-Y translational stage) is adjusted to 25×25 mm (Fig. S20). Sol spin coating can be carried out on this 20×20 mm substrate. We have added the description in the Materials and Methods section of the revised manuscript on line 325-329, “The films are built by alternately depositing of ink in the X and Y directions. To form highly dense films, we set the distance between the neighbouring parallel paths of substrate to 4 mm to ensure partial overlap between deposited materials (Fig.S20). During the deposition of films and patterns, it is inevitable to enlarge the range of ejected droplets to undesired region for the uniformity of the films on the desired substrates.” and line 332-334, “Sol-infiltration on every five layers of the as-deposited films by means of spin-coating (1000 rpm for 30s) is performed. The films are then dried at 200 °C for 2 min to remove the organic components and reduce internal stress.”

Figure S20. Schematic demonstration of the X-Y translational stage movement paths for building the films and patterns.

4. The proposed EDP process enables the fabrication of large-scale and fast production of piezoelectric devices. However, electrostatic deposition using dielectric masks generally enabled to fabricate micro-patterns, but also deposition of used materials occurred on the dielectric mask, where unused areas. How about the fabrication efficiency of the products in terms of the material usage? The following articles are relevant to your work, which are suggested to be considered: 10.1016/j.ceramint.2020.07.249, 10.1038/s41598-021-01043-6.

Response:

We thank the referee for the careful review and professional comments. Given that the mask will be detached after the deposition and heat treatment, it is believed that a significant amount of unused deposited materials would be wasted. We calculated the coverage ratio of PZT on the target region and mask region by measuring the covered area of PZT based on ImageJ. Using polyimide mask with ~ 200 μm -wide grooves, the coverage ratio of PZT on the target region and mask region is 92.5 ± 1.2 % and 21.6 ± 6.6 %, respectively. A sparse amount of PZT is deposited on the mask region due to the unstable jet and the electrostatic repulsion between the deposited and coming droplets. We have added the description in the revised manuscript on line 168-173, “Given that the mask will be detached after the deposition and heat treatment, it is believed that a significant amount of unused deposited materials would be wasted. For example, using polyimide mask with ~ 200 μm -wide grooves, the coverage ratio of PZT on the target region and mask region is 92.5 ± 1.2 % and 21.6 ± 6.6 %, respectively. A sparse amount of PZT is deposited on the mask region due to the unstable jet and the electrostatic repulsion between the deposited and coming droplets.” and line 339-340, “We calculated the coverage ratio of PZT on the target region and mask region by measuring the covered area of PZT via ImageJ.”

5. In the Abstract, the authors declared that the EDP process allows for one-step fabrication of PZT nanoparticles, films, and micro-patterns. However, spin-coating with PZT sol conducted to further improve the density of the as-deposited films. Without the spin-coating process, can PZT films and micro-patterns be fabricated well? If not, the word ‘one-step’ should be applied to fabricate only PZT nanoparticles in the whole manuscript or removed to avoid misunderstanding to readers.

Response:

We thank the referee for the careful review and professional comments. The purpose of spin-coating with PZT sol is to further improve the density of the as-deposited films, and thus

improve their electrical properties. The calculation of depositing speed of EDP and the piezoresponse measurements of films are based on PZT films with sol spin-coating. In order to avoid misunderstanding to readers, the word 'one-step' has been removed in the whole manuscript.

6. In general, the performance of energy harvesters includes not only output voltage but also output current. However, in the paper, only the output voltage performance graph is showed. Therefore, to show the device performance, the current graph according to variable, such as frequency and force, is recommended to be added.

Response:

We thank the referee for the careful review and professional comments. The output current of the HIPCT-based energy harvester as a function of the compressive pressure/resistance are measured, as shown in Fig. S15e, f. The output current increases gradually from $\sim 15 \mu\text{A}$ to $\sim 75 \mu\text{A}$ with the increase of the compressive pressure, while the output current decreases gradually from $\sim 60 \mu\text{A}$ to $\sim 1 \mu\text{A}$ with the increase of the resistance. We have added the description in the revised manuscript on line 258-262, “In addition, the output current of the HIPCT-based energy harvester as a function of the compressive pressure/resistance are measured, as shown in Fig. S15e, f. The output current increases gradually from $\sim 15 \mu\text{A}$ to $\sim 75 \mu\text{A}$ with the increase of the compressive pressure, while the output current decreases gradually from $\sim 60 \mu\text{A}$ to $\sim 1 \mu\text{A}$ with the increase of the resistance.”

Fig. S15. Optical photograph of piezocomposite energy harvester (a). The output voltages of the energy harvester under different loading frequency with same compress pressure of 2.2 MPa (b). The output voltages of the energy harvester under different compressive pressure with same loading frequency of 30 Hz. Its sensitivity is ~ 13.6 V/MPa and R-squared is 0.978 (c). Voltage responses during 28000 cycles under the compressive pressure of 1.8 MPa (d). Short-circuit current signals of the energy harvester under different compressive pressure with same loading frequency of 30 Hz (e). Output current depend on the load impedance of 2.2 MPa compressive pressure with a frequency of 30 Hz (f).

7. In general, the performance of a pressure sensor is verified by sensitivity, such as the amount of generated power (V/N or V/Pa) according to the pressure (force) and R-squared score. However, in Figure S11, which shows the performance as the pressure sensor, it is difficult to know quantitatively the sensitivity of the device. Correspondingly, it is recommended to add the above two indicators for the quantitative inspection.

Response:

We thank the referee for the careful review and professional comments. The amount of generated power according to the pressure is 13.6 ± 0.5 V/MPa, and R-squared score is 0.978. We have added these description in Fig. S15c.

Fig. S15. Optical photograph of piezocomposite energy harvester (a). The output voltages of the energy harvester under different loading frequency with same compress pressure of 2.2 MPa (b). The output voltages of the energy harvester under different compressive pressure with same loading frequency of 30 Hz. Its sensitivity is ~ 13.6 V/MPa and R-squared is 0.978 (c). Voltage responses during 28000 cycles under the compressive pressure of 1.8 MPa (d). Short-circuit current signals of the energy harvester under different compressive pressure with same loading frequency of 30 Hz (e). Output current depend on the load impedance of 2.2 MPa compressive pressure with a frequency of 30 Hz (f).

8. The Figure 3e is not match the manuscript. Also, some typos and grammar error are confirmed in the manuscript and figures. The authors should be corrected throughout the text and figures.

Response:

We thank the referee for the careful review and professional comments. The description of Figure 3e has revised in the revised manuscript on line 174-177, “**Patterned structures can be deposited on flexible planar substrates and 3D free-form substrates (Fig. 3e), demonstrating the aesthetic diversity of the EDP patterns. The printed letters ‘Hello world’ composed of different materials (silver and PZT) (Fig. 3e and Fig. S9a) demonstrate the material versatility of EDP.**”

The typos and grammar error have been corrected in the revised manuscript and figures.

Reviewer #3 (Remarks to the Author):

The work entitled “Ultrafast and Versatile Electrostatic Disc Microprinting for Piezoelectric Elements” submitted by Li, Zhang, et al. is very innovative and without doubt of high interest for the scientific community in the field of functional materials and related technologies.

However, at least in my view, the paper shows some weaknesses in detail that should be addressed by the authors prior to publication.

The reported work in the main deals with deposition technology, and in my role as a technologist I miss some concrete information about the presented approach compared to other work in the field.

One of the core features of the proposed technology, as far as I understand, is the self-polarizing feature of the fabricated films and patterns. In that respect I miss the classification in comparison with other work in this field, e.g., the work of Khan et al. recently published in 2022 [Khan, Rukhsar Ali, et al. "Development of self-polarized PVDF films on carbon fabrics for sensing applications." *The Journal of The Textile Institute* 113.10 (2022): 2208-2214].

Response:

We thank the referee for the careful review and professional comments. A common strategy to enhance piezoelectric coefficient of the inorganic piezoelectric films is to construct phase boundaries by tuning the composition chemically. However, the dopant content is only a few percent and often involves highly volatile elements (Pb, K, Na, and Bi), which hinders obtaining reliable and reproducible synthesis of films. Electrical poling can cause domain motions which are responsible for a deformation of the piezoelectric films and are the source of the piezoelectric effect. In our work, we demonstrate that during EDP process, the in-situ electric field triggers the structure distortion and contributes to an outstanding effective piezoelectric coefficient of $\sim 590 \text{ pm V}^{-1}$. It is noteworthy that our PZT films need be polarized at $130 \text{ }^\circ\text{C}$ under 10 kV mm^{-1} for 30 min.

Among all the stable crystalline variants (α , β , γ , and δ phases) for the PVDF polymers, the polar β -phase shows the best piezoelectric properties. The large piezoelectricity of the β -phase originate from the all-trans planar zigzag conformation with its aligned dipoles perpendicular to the c-axis. The crystallinity, β -phase fraction, and dipole orientation are critical to the overall piezoelectric performances of the PVDF polymers. Different strategies, such as thermodynamically biaxial stretching, pressing and folding technique, solvent vapor annealing, melt spinning, and crystallization by solution have been developed to improve the crystallinity

and obtain the self-polarized β -phase PVDF. Khan et al. deposited self-polarized PVDF films on the carbon fabric through solution cast technique. [Khan, Rukhsar Ali, et al. "Development of self-polarized PVDF films on carbon fabrics for sensing applications." The Journal of The Textile Institute 113.10 (2022): 2208-2214].

Our EDP process can trigger structural distortions of PZT films (Table S5) and contributes to the improvement of ferroelectric and piezoelectric properties. Electrical poling is one of the necessary steps for our PZT films to exhibit piezoelectricity. We have added these descriptions in Results section of the revised manuscript on line 207-212, “The crystallographic refinement results further confirm that MPB PZT films composed of R phase and T phase are synthesized (Fig. S12). Their lattice parameters are summarized in Table S5. Compared with those of the dip-coated PZT films, all lattice constants (a_R , a_T , and c_T) of EDP PZT films have increased, and the level of c_T/a_T increase. The rhombohedral distortion, $90^\circ - \alpha_R$ also increases from 0.242° to 0.351° . These structural distortions (Fig. 4c) are caused by in-situ electric field^{39,40}, and will contribute the improvement of ferroelectric and piezoelectric properties⁴¹.” line 229-234, “A common technique to enhance piezoelectric properties of films is to construct phase boundaries by tuning the composition chemically. However, the dopant content is only a few percent and often involves highly volatile elements (Pb, K, Na, and Bi), which hinders obtaining reliable and reproducible synthesis of films. Our strategy of electric field-induced structure distortion shows excellent potential for improving structural properties to get high-performance piezoelectric films.” and Materials and Methods section on line 341-343, “Sintering treatment at 800°C for 30 min is performed for synthesizing perovskite structures. Polarization of PZT films is conducted at 130°C under 10 kV mm^{-1} for 30 min.”

Table S5. Lattice parameters of the EDP deposited PZT films and the dip-coated PZT films, at room temperature.

Film Type	Phase fraction (%)		Lattice parameter				
	Tetragonal	Rhombohedral	a_R (Å)	α_R ($^\circ$)	a_T (Å)	c_T (Å)	c/a
With electric field	50.336	49.664	4.082	89.649	4.036	4.124	1.0218
Without electric field	49.257	50.743	4.077	89.758	4.040	4.117	1.0192

Besides, when discussing compatibility of common patterning techniques with flexible and stretchable substrates, the authors should mention PVDF and related co-polymers more

explicitly in the state of the art, since these materials seem increasingly important in the relevant field.

Response:

We thank the referee for the careful review and professional comments. Poly(vinylidene fluoride) (PVDF) is a very promising material for fabricating flexible and wearable devices due to its ferroelectricity, piezoelectricity, flexibility as well as excellent. Micropatterning of PVDF films is an essential step in fabrication process in order for the sample to be integrated in microelectromechanical systems. Different techniques have been used to pattern PVDF. One conventional technique is the combination of spin coating and reactive ion etching (RIE)^{1,2}: firstly, RIE is used to define the patterns on the substrates. Then, spinning coating allows depositing a thin layer of PVDF. Finally lift-off is realized to remove the PVDF deposited on the mask. Two types of RIE masks are used to define PVDF patterns on the substrate. The first one consisted of a lithographically defined resist pattern using conventional photoresist SU8. The second type of etch mask is a shadow mask of PDMS made by soft lithography. Some techniques from classical micro-fabrication have also been used for patterning PVDF: laser micromachining³, O₂ dry etching⁴, and selective shear patterning⁵. In opposite of the technique evoked above which need special equipment and are low throughput, soft lithography⁶ called hot embossing or micro-imprinting is developed for PVDF patterning: the patterns are transferred from a mold onto a thermoplastic polymer by heating this polymer above its glass transition temperature. When heated, the polymer is softened which can then be molded into the mold structured with the help of a pressure applied between the mold and the substrate. We have added these descriptions in Introduction section of the revised manuscript on line 34-39, “Conventional micro-patterning techniques rely on screen printing¹⁵ and photolithography/chemical etching^{16,17}. However, these techniques often require high sintering temperatures (> 1000°C), complex and time-consuming processing conditions, and hazardous materials. For example, patterning of poly(vinylidene fluoride) (PVDF) films needs the combination of spin coating and reactive ion etching^{18,19} or micromold-assisted process²⁰ (soft lithography). For inorganic piezoelectric ceramics, the above patterning techniques lack compatibility with flexible and stretchable substrates.”

1. Zabek, D., Taylor, J., Boulbar, E. Le & Bowen, C. R. Micropatterning of Flexible and Free Standing Polyvinylidene Difluoride (PVDF) Films for Enhanced Pyroelectric Energy Transformation. *Adv. Energy Mater.* **5**, 1401891 (2015).
2. Fadzallah, I. A. *et al.* Micropatterning and Integration of Electrospun PVDF Membrane Into Microdevice. *J. Microelectromechanical Syst.* **29**, 438–445 (2020).

3. Bartnik, A., et al. Efficient micromachining of poly (vinylidene fluoride) using a laser-plasma EUV source. *Applied Physics A* **106**, 551-555 (2012).
4. Miki, Hirofumi, et al. Fabrication of microstructure array directly on β -phase poly (vinylidene fluoride) thin film by O₂ reactive ion etching. *Journal of Micromechanics and Microengineering* **25.3**, 035026 (2015).
5. Chang, Jiyoun, et al. One-step micropatterning of highly-ordered semi-crystalline poly (vinylidene fluoride-co-trifluoroethylene) films by a selective shear and detachment process. *Organic Electronics* **12.1**, 98-107 (2011).
6. Choi, Y.-Y. *et al.* Vertically aligned P(VDF-TrFE) core-shell structures on flexible pillar arrays. *Sci. Rep.* **5**, 10728 (2015).
15. Chen, Y. *et al.* PZT ceramics fabricated based on stereolithography for an ultrasound transducer array application. *Ceram. Int.* **44**, 22725–22730 (2018).
16. Wang, Z. *et al.* Manipulation of charge transfer and transport in plasmonic-ferroelectric hybrids for photoelectrochemical applications. *Nat. Commun.* **7**, 10348 (2016).
17. Dagdeviren, C. *et al.* Conformable amplified lead zirconate titanate sensors with enhanced piezoelectric response for cutaneous pressure monitoring. *Nat. Commun.* **5**, 4496 (2014).
18. Zabek, D., Taylor, J., Boulbar, E. Le & Bowen, C. R. Micropatterning of Flexible and Free Standing Polyvinylidene Difluoride (PVDF) Films for Enhanced Pyroelectric Energy Transformation. *Adv. Energy Mater.* **5**, 1401891 (2015).
19. Fadzallah, I. A. *et al.* Micropatterning and Integration of Electrospun PVDF Membrane Into Microdevice. *J. Microelectromechanical Syst.* **29**, 438–445 (2020).
20. Choi, Y.-Y. *et al.* Vertically aligned P(VDF-TrFE) core-shell structures on flexible pillar arrays. *Sci. Rep.* **5**, 10728 (2015).

On page 5 (line 209) the authors claim, that their technology facilitates the realization of “high-quality” piezoelectric films. However, the label “high-quality” of thin films comprises more than a high piezoelectric coefficient, it is also concerned with things such as surface roughness, achievable layer uniformity and homogeneity, and the constraints with regard to substrate dimensions that arise as a consequence.

Response:

We thank the referee for the careful review and professional comments. This vague expression has been deleted, and we have added these descriptions in Results section of the revised manuscript on line 227-229, “EDP strategy and in-situ electrostatically crystallographic structure upgrading are capable of fabricating PZT films with the thickness up to 50 μm and effective piezoelectric coefficient of $\sim 560 \text{ pm/V}$.”

Authors should also comment on issues like reproducibility (piezoelectric and structural properties) in fabrication, as well as on durability and aging effects of the fabricated PZT

structures. For example, a well-known problem of PZT is resistance degradation under electrical field load, additionally strongly dependent on the used electrode material. How do the layers fabricated by the proposed technology perform in this respect?

Response:

We thank the referee for the careful review and professional comments. As for all printing techniques, the process parameters should be optimized for the given ink and the reproducibility of the printed products is of principal importance. Here, the reproducibility and consistency of the EDP process is assessed by analyzing the piezoelectric and structural properties of PZT films/patterned lines that are printed via the optimized process parameters (Table S2). Three PZT films with the thickness of $<10\ \mu\text{m}$ are deposited at different times. PFM is used for piezoresponse measurements using an Asylum Cypher ES AFM system with a conductive probe (Nano world Arrow-EFM). For calculating piezoelectric coefficient d_{33} , an area ($500 \times 500\ \text{nm}$) was scanned in DART (dual AC resonance tracking) mode with varied tip drive voltage (from 10 mV to 200 mV), and the corresponding out-of-plane piezoelectric amplitudes are recorded over the scanned area. The maximum and minimum measured value is ~ 470 and $\sim 648\ \text{pm/V}$, respectively. The mean value is $\sim 560\ \text{pm/V}$. Three samples with PZT linear structures arrays are fabricated via polyimide mask with $\sim 100\ \mu\text{m}$ -wide grooves. The widths of printed lines for each printed sample are measured at ten different positions. There is no statistically significant difference among any of the widths measured at the same sample (Fig. S21). These data indicate that our EDP can fabricate PZT films and micropatterns in a consistently uniform manner. We have added these descriptions in the revised manuscript (Results Versatility and reproducibility of the EDP process) on line 273-287, “As for all printing techniques, the process parameters should be optimized for the given ink and the reproducibility of the printed products is of principal importance. Here, the reproducibility and consistency of the EDP process is assessed by analyzing the piezoelectric and structural properties of PZT films/patterned lines that are printed via the optimized process parameters (Table S2). Three PZT films with the thickness of $<10\ \mu\text{m}$ are deposited at different times. PFM is used for piezoresponse measurements using an Asylum Cypher ES AFM system with a conductive probe (Nano world Arrow-EFM). For calculating piezoelectric coefficient d_{33} , an area ($500 \times 500\ \text{nm}$) was scanned in DART (dual AC resonance tracking) mode with varied tip drive voltage (from 10 mV to 200 mV), and the corresponding out-of-plane piezoelectric amplitudes are recorded over the scanned area. The maximum and minimum measured value is ~ 470 and $\sim 648\ \text{pm/V}$, respectively. The mean value is $\sim 560\ \text{pm/V}$. Three samples with PZT linear structures arrays are fabricated via polyimide mask with $\sim 100\ \mu\text{m}$ -wide grooves. The widths of printed lines for each printed sample are measured at ten different positions. There

is no statistically significant difference among any of the widths measured at the same sample (Fig. S21). These data indicate that our EDP can fabricate PZT films and micropatterns in a consistently uniform manner.”

The effective piezoelectric coefficient value of PZT films has been corrected to ‘560 pm V⁻¹’ in the whole manuscript.

Normally, the long-time service process of the piezoelectric films is accompanied with the increase of the leakage current, eventually resulting in the breakdown. The above phenomenon is called as the electric degradation. Here, the leakage current measurements of our PZT films with Ag electrodes is conducted under a DC bias field of 150 kV/cm and a temperatures of 180 °C. The applied electric field is directed from the top to the bottom electrode. Ag top electrodes with diameters of 1 mm are printed onto the PZT surface by EDP (Fig. S22a). Measurements of current are made 60 s after any change in value to allow the current to stabilize. Fig. S22b presents the characteristic leakage current response from our PZT film. The steady-state current rises gradually upon resistance degradation. We have added these descriptions in the revised Supplementary Information (Note S3 Leakage current characteristics of EDP PZT films) on line 98-106, “Normally, the long-time service process of the piezoelectric films is accompanied with the increase of the leakage current, eventually resulting in the breakdown. The above phenomenon is called as the electric degradation. Here, the leakage current measurements of our PZT films with Ag electrodes is conducted under a DC bias field of 150 kV/cm and a temperatures of 180 °C. The applied electric field is directed from the top to the bottom electrode. Ag top electrodes with diameters of 1 mm are printed onto the PZT surface by EDP (Fig. S22a). Measurements of current are made 60 s after any change in value to allow the current to stabilize. Fig. S22b presents the characteristic leakage current response from our PZT film. The steady-state current rises gradually upon resistance degradation.”

Table S2. Optimized process parameters for EDP depositing PZT films/patterns.

Parameter	Value
Ink concentration (mass ratio of particles to sol)	30-50 wt.%
Depositing speed (mm s^{-1})	5~30
Distance between disc to substrate (mm)	2~8
Supply rate of ink ($\mu\text{l min}^{-1}$)	3-200
Applied voltage (kV)	3.0~10.0

Figure S21. Distribution of measured line widths for 3 different samples at ten positions (blue dots). No significant differences exist between the width distributions of any of the different positions. Distribution of measured d_{33} for 3 different samples at two positions (pink dots). The tops and bottoms of the dashed line represent the maximum measured value and the minimum measured value, respectively.

Figure S22. Schematic diagram of circuit used for electrical degradation measurement (a). Variation in

leakage current with time for EDP PZT films (b). The measurement is conducted at 180 °C with a DC field of 150 kV/cm.

REVIEWERS' COMMENTS

Reviewer #1 (Remarks to the Author):

The authors successfully addressed, step-by-step, all the comments raised on the former version of this manuscript. As a consequence, the manuscript reads now much better, the logic of the experiments is easier to follow, and the experimental details are easier to find. The rewriting of some of the sections of the manuscript will provide the readers with a clear idea of the limitations and the range of applicability of the "electrostatic disc microprinting" technique, facilitating its adoption. Therefore, I recommend publishing this manuscript in its current form.

Reviewer #2 (Remarks to the Author):

In this revised manuscript, the authors developed ultrafast and versatile electrostatic disk microprinting (EDP) for piezoelectric elements. The authors conducted electrical simulation and experimental validation to investigate the suggested disk EDP process. Furthermore, additional investigations for the deposit behavior of the disk EDP process with fewer tips were conducted and explained clearly, given the 3D conformal fabrication process. Also, the fabricated piezo device showed outstanding electrical performance in various pressure conditions as an energy harvester and self-powered sensor. Considering that the revised manuscript provided a versatile deposit process, concrete analysis of the process, and high performance of its application, the manuscript should be considered to be accepted.

Reviewer #3 (Remarks to the Author):

The authors of the submitted article entitled „Ultrafast and Versatile Electrostatic Disc Microprinting for Piezoelectric Elements“ have properly responded to all of my issues raised in the first review.

Besides that, I have not noticed any further points of criticism when reading the revised manuscript. Thus, I would recommend publication in its present form.

Thank you for your review and constructive comments on the manuscript (NCOMMS-23-19123A) entitled "*Ultrafast and Versatile Electrostatic Disc Microprinting for Piezoelectric Elements*" submitted for publication on Nature Communications. We have revised the manuscript carefully. Please find responses to reviewers' comments below.

Reviewer Comments (and change made in accordance)

Reviewer #1 (Remarks to the Author):

The authors successfully addressed, step-by-step, all the comments raised on the former version of this manuscript. As a consequence, the manuscript reads now much better, the logic of the experiments is easier to follow, and the experimental details are easier to find. The rewriting of some of the sections of the manuscript will provide the readers with a clear idea of the limitations and the range of applicability of the "electrostatic disc microprinting" technique, facilitating its adoption. Therefore, I recommend publishing this manuscript in its current form.

Response:

We are grateful for the reviewer's affirmation on our revised manuscript. Thank you for recommending our revised manuscript to be published in Nature Communications.

Reviewer #2 (Remarks to the Author):

In this revised manuscript, the authors developed ultrafast and versatile electrostatic disk microprinting (EDP) for piezoelectric elements. The authors conducted electrical simulation and experimental validation to investigate the suggested disk EDP process. Furthermore, additional investigations for the deposit behavior of the disk EDP process with fewer tips were conducted and explained clearly, given the 3D conformal fabrication process. Also, the fabricated piezo device showed outstanding electrical performance in various pressure conditions as an energy harvester and self-powered sensor. Considering that the revised manuscript provided a versatile deposit process, concrete analysis of the process, and high performance of its application, the manuscript should be considered to be accepted.

Response:

We are grateful for the reviewer's affirmation on our revised manuscript.

Reviewer #3 (Remarks to the Author):

The authors of the submitted article entitled "Ultrafast and Versatile Electrostatic Disc Microprinting for Piezoelectric Elements" have properly responded to all of my issues raised in the first review.

Besides that, I have not noticed any further points of criticism when reading the revised manuscript. Thus, I would recommend publication in its present form.

Response:

We are grateful for the reviewer's affirmation on our revised manuscript. Thank you for recommending our revised manuscript to be published in Nature Communications.